# Efficient and robust estimation of many-qubit Hamiltonians

Daniel Stilck França [1,2] ✉, Liubov A. Markovich[3,4], V. V. Dobrovitski[3], Albert H. Werner [1,5] & Johannes Borregaard[3,6]

Characterizing the interactions and dynamics of quantum mechanical systems is an essential task in developing quantum technologies. We propose an efficient protocol based on the estimation of the time-derivatives of few qubit observables using polynomial interpolation for characterizing the underlying Hamiltonian dynamics and Markovian noise of a multi-qubit device. For finite range dynamics, our protocol exponentially relaxes the necessary time-resolution of the measurements and quadratically reduces the overall sample complexity compared to previous approaches. Furthermore, we show that our protocol can characterize the dynamics of systems with algebraically decaying interactions. The implementation of the protocol requires only the preparation of product states and single-qubit measurements. Furthermore, we improve a shadow tomography method for quantum channels that is of independent interest and discuss the robustness of the protocol to various errors. This protocol can be used to parallelize the learning of the Hamiltonian, rendering it applicable for the characterization of both current and future quantum devices.

Large quantum devices consisting of tens to hundreds of qubits have been realized across various hardware architectures[1–4] representing a significant step towards the realization of quantum computers and simulators with the potential to solve outstanding problems intractable for classical computers[5,6]. However, continued progress towards this goal requires careful characterization of the underlying Hamiltonians and dissipative dynamics of the hardware to mitigate errors and engineer the desired dynamics. The exponential growth of the dimension of the state space of a quantum device with the number of qubits renders this an outstanding challenge broadly referred to as the Hamiltonian learning problem[7–35].

To tackle this challenge, previous approaches make strong assumptions such as the existence of a trusted quantum simulator capable of simulating the unknown Hamiltonian[20,21] or the capability of preparing particular states of the Hamiltonian such as steady states

and Gibbs states[23,25,26,29,36,37], which may be difficult for realistic devices subject to various decoherence mechanisms.

Alternatively, several works[30–32] are built on the observation that a Master equation describes the evolution of any system governed by Markovian dynamics. Through this, one obtains a simple linear relation between time derivatives of expectation values and the parameters of the Hamiltonian, jump operators and decay rates (jointly referred to as the parameters of the Lindbladian $\mathcal{L}$) governing the system. Furthermore, for finite range interactions, these approaches can estimate the parameters of the Lindbladian to a given precision from a number of samples that is independent of the system's size[30–32].

A significant drawback of these approaches is that the time derivatives are estimated using finite difference methods. Obtaining a good precision thus requires high time resolution, which is experimentally challenging given the finite operation time of gates and

[1]QMATH, Department of Mathematical Sciences, University of Copenhagen, Universitetsparken 5, 2100 Copenhagen, Denmark. [2]Univ Lyon, ENS Lyon, UCBL, CNRS, Inria, LIP, F-69342, Lyon Cedex 07, France. [3]QuTech and Kavli Institute of Nanoscience, Delft University of Technology, Delft 2628 CJ, The Netherlands. [4]Instituut-Lorentz, Universiteit Leiden, P.O. Box 9506, Leiden 2300 RA, The Netherlands. [5]NBIA, Niels Bohr Institute, University of Copenhagen, Blegdamsvej 17, 2100 Copenhagen, Denmark. [6]Department of Physics, Harvard University, Cambridge, MA 02138, USA. ✉e-mail: daniel.stilck_franca@ens-lyon.fr

measurements. To estimate a Lindbladian parameter up to an additive error $\epsilon$, the system has to be probed at times $\mathcal{O}(\epsilon)$ apart and expectation values of observables have to be estimated up to a precision of $\mathcal{O}(\epsilon^2)$, which translates to an overall $\mathcal{O}(\epsilon^{-4})$ sample complexity to estimate each parameter.

In this article, we propose a protocol that alleviates these daunting experimental requirements. Our protocol requires only a time resolution of $\mathcal{O}(\text{polylog}(\epsilon^{-1}))$ representing an exponential improvement compared to previous protocols and gives an overall sample complexity to recover *all* parameters of a *k*-local *n* qubit Lindbladian up to precision $\epsilon$ of $\mathcal{O}(9^k \epsilon^{-2} \text{polylog}(n, \epsilon^{-1}))$. We obtain this by estimating time derivatives using multiple temporal sampling points and robust polynomial interpolation[38]. Furthermore, we show how to use shadow process tomography methods to estimate multiple parameters in parallel. In particular, we improve the results of refs. 39, 40 in extending the framework of classical shadows to processes and Pauli matrices with an alternative proof, a result that is of independent interest. We also extend our analysis to long-range (algebraically decaying) interactions in the systems, obtaining the first results for such systems to the best of our knowledge. The necessary operations for our protocol are measurements in the Pauli bases on time-evolved product states consisting of Pauli eigenstates. These minimal requirements make our protocol feasible for characterization of both current and future quantum devices.

## Results

In order to use our protocol for an efficient characterization of a quantum device, two assumptions should be fulfilled:

1. The quantum device implements an (unknown) Markovian quantum evolution on *n* qubits described by a time-independent Lindbladian, $\mathcal{L}$.
2. We assume knowledge of the general structure of the interaction graph of the device i.e., which qubits are coupled to each other. Importantly, no assumptions are made regarding the couplings' exact form.

The first assumption ensures that the evolution of a general observable, $O(t)$ is described by the Master equation, i.e., $\frac{d}{dt}O(t) = \mathcal{L}(O(t))$. We note that the Lindbladian captures both the Hamiltonian evolution and the dissipative dynamics of the device.

The second assumption bounds the size of the estimation task. It corresponds to making some assumptions about the locality of the generator, as evolutions without some locality assumptions have exponentially many parameters. However, having prior knowledge that, e.g., nearest neighbor couplings dominate in the device, makes the estimation task tractable. For now, we will assume that we know which qubits interact. Later, we will show that a bound on the support of each interaction and some technical assumptions on the evolution suffice to also learn the interaction graph from data.

Using the knowledge of the interaction graph, one can expand the Lindbladian in an operator basis, $\{P_i\}$ constructed from tensor products of single-qubit Pauli matrices and the identity[41]:

$$\mathcal{L}(O) = \sum a_i[P_i, \rho] + \sum D_{i,j}\left(P_i^\dagger O P_j - \frac{1}{2}\{P_j^\dagger P_i, O\}\right) \tag{1}$$

Such an expansion is always possible since this basis amounts to a Hilbert-Schmidt orthogonal set of Hermitian operators spanning the entire vector space. Note that the coefficient matrix $D_{i,j}$ needs to be positive for it to form a valid Lindbladian. Estimating the set of expansion coefficients $\{a_i, D_{i,j}\}$ gives an estimation of $\mathcal{L}$ and thus a full characterization of the system.

It is well known that the Master equation for the time derivative of the expectation value of a local observable $O$ at time $t = 0$ for a given initial state $\rho$ of the system gives us a linear equation for the expansion coefficients[30–32]. We use this to estimate the expansion coefficients going through three stages of *classical pre-processing*, *quantum processing*, and *classical post-processing* (see Fig. 1).

### Classical pre-processing

After expanding $\mathcal{L}$ in an operator basis, the following steps are performed.

1. Find a suitable complete set, $\{(\rho_j, O_i)\}$ of multi-qubit product states ($\rho_j$) and observables ($O_i$) for which the Master equation involves only a few selected expansion parameters of the Lindbladian for each element of the set. The set is complete in the sense that all expansion coefficients can be found by solving the Master equations for all elements in the set. As we show below, such a set can readily be found by considering initial states where only a few qubits are initialized as different eigenstates of the Pauli matrices while the remaining qubits are prepared in the maximally mixed state $I/2$.
2. Calculate the expectation values appearing on the right-hand side of the Master equations $\frac{d}{dt}\text{tr}[\rho_j O_i(t)] = \text{tr}[\rho_j \mathcal{L}(O_i)]$ for all elements in the set $\{(\rho_j, O_i)\}$ in terms of the expansion coefficients. Since the initial states and the observables are products, this can be done efficiently.

### Quantum processing

In order to solve for the expansion coefficients, we also need the values of the time-derivatives appearing on the left-hand side of the Master equations, i.e., $\frac{d}{dt}\text{tr}[\rho_j O_i]$. These can be estimated using the quantum device. The naïve approach is the following:

1. The quantum device is prepared in initial state $\rho_j$ and evolved for a time $t_k \in \{t_0, t_1, \ldots, t_{\max}\}$ after which the observable $O_i$ is measured.
2. The above procedure is repeated for each element in the set $\{(\rho_j, O_i)\}$ for all evolution times $t_k$ to obtain estimates of $\langle O_i(t_k)\rangle_j = \text{tr}[\rho_j(t_k)O_i]$ where $\rho_j(t_k)$ is the state of the system having evolved for time $t_k$ from the initial state $\rho_j$. We note that the single qubit mixed states can be simulated by sampling eigenstates of the Pauli matrices at random. For this naïve approach, the sample complexity increases linearly with the size of the set $\{(\rho_j, O_i)\}$ since the expectation values $\langle O_i(t_k)\rangle_j$ are estimated sequentially. However, we also propose a variation of the classical shadows protocol of ref. 42 for process tomography that can reduce this to a logarithmic scaling. In essence, we can obtain estimates of all elements in the set $\{(\rho_j, O_i)\}$ in parallel. This is done by the following steps:
1. Every qubit is prepared in a random Pauli matrix eigenstate and the system is evolved for a time $t_k$ after which each qubit is measured in a random single-qubit Pauli basis.
2. The above procedure is repeated $O(3^{w_1 + w_2} \epsilon^{-2} \log(K))$ times set by the required precision, $\epsilon$, of the estimates, the size, $K$, of the set $\{(\rho_j, O_i)\}$ and the weights (i.e., maximum number of sites differing from identity) of $\rho_j$ ($w_1$) and $O_i$ ($w_2$). The whole procedure is repeated for all times $t_k \in \{t_0, t_1, \ldots, t_{\max}\}$. From the measurement statistics of the above procedure, it is possible to obtain accurate estimates of all $\langle O_i(t_k)\rangle_j$ and thus a parallel estimation is possible. We refer to Sec. V and the Supplementary Note for more details and proof of the method.

Whether to use the sequential approach or the parallel approach depends on the number of qubits and the weight of the states and observables in the set $\{(\rho_j, O_i)\}$. For few qubit processors, the sequential protocol may require fewest samples, however for local Hamiltonians on a lattice, the logarithmic scaling in system size of the parallel method will quickly be advantageous for larger

## 1. Classical pre-processing

Determine interaction graph

Expansion of Lindbladian

Set of initial product states and single qubit

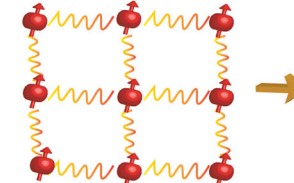

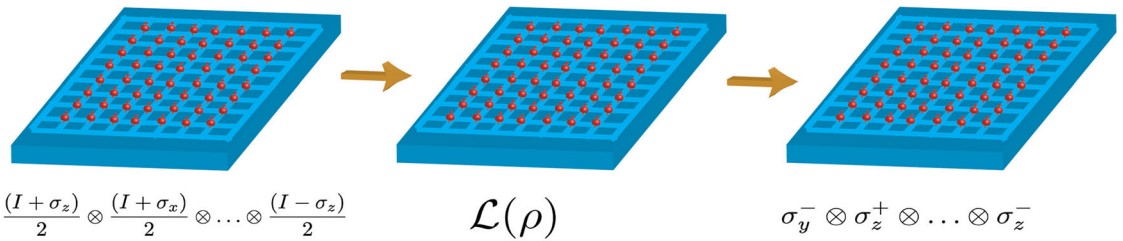

$$\mathcal{L}(\rho) = \sum a_i[P_i, \rho]$$
$$+ \sum D_{i,j}(P_i \rho P_j^\dagger - \frac{1}{2}\{P_j^\dagger P_i, \rho\})$$

$$\{(\rho_j, O_i)\}$$

$$\frac{d}{dt}\mathrm{tr}\,[\rho_j O_i] = \mathrm{tr}\,[\rho_j \mathcal{L}(O_i)]$$

## 2. Quantum processing

Prepare product states of single qubit Pauli states

Evolve for a number of time steps

Measure in single qubit Pauli bases

$$\frac{(I + \sigma_z)}{2} \otimes \frac{(I + \sigma_x)}{2} \otimes \ldots \otimes \frac{(I - \sigma_z)}{2}$$

$$\mathcal{L}(\rho)$$

$$\sigma_y^- \otimes \sigma_z^+ \otimes \ldots \otimes \sigma_z^-$$

## 3. Classical post-processing

Fit time traces of observables to low degree polynomials,

Solve set of linear equations to estimate the Lindbladian

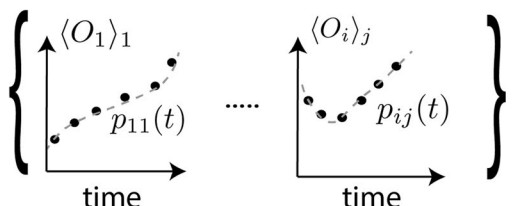

$$\left\{\frac{d}{dt}p_{ij}(t) \approx \mathrm{tr}\,[\rho_j \mathcal{L}(O_i)]\right\}$$

$$\mathcal{L}_{est} \approx \mathcal{L}$$

**Fig. 1 | Sketch of the proposed protocol to estimate an unknown Lindbladian, $\mathcal{L}$, of a multi-qubit device.** In the first step of classical pre-processing, the interaction graph between qubits is identified from the physical connectivity of the device. Then the unknown Lindbladian is written in a general form using an operator basis of Pauli strings, $\{P_i\}$ and a suitable set of initial states and observables, $\{(\rho_j, O_i)\}$ is chosen. In the second step of quantum processing, a time trace (expectation value) of each element of the set is obtained from the preparation and evolution of single qubit Pauli eigenstates on the quantum device followed by measurements in single qubit Pauli basis. In the last step of classical post-processing, each time trace is fitted to a low-degree polynomial to estimate the derivative of the observable. From these, an estimate of the Lindbladian, $\mathcal{L}_{est}$, is obtained from the Master equation.

processors. Importantly, both methods only require the preparation of single qubit Pauli states and measurements in single-qubit Pauli bases.

### Classical post-processing

The final part of the characterization involves estimating $\frac{d}{dt}\mathrm{tr}[\rho_j O_i(t)]$ from the experimentally obtained time trace of $\langle O_i(t_k)\rangle_j$ and solving for the expansion coefficients $\{a_i\}$. This involves

1. Fit the time trace of $\langle O_i(t_k)\rangle_j$ with a low-degree polynomial in the time, $p_{ij}(t)$ and estimate $\frac{d}{dt}\mathrm{tr}[\rho_j O_i]$ as $\frac{d}{dt}p_{ij}(t)|_{t=0}$. This is done for each element in the set $\{(\rho_j, O_i)\}$.

2. Solve the set of linear equations from the Master equations $\frac{d}{dt}\mathrm{tr}[\rho_j O_i] = \mathrm{tr}[\rho_j \mathcal{L}(O_i)]$ with respect to the expansion coefficients. This is possible since $\frac{d}{dt}\mathrm{tr}[\rho_j O_i]$ has been estimated from the polynomial fits and all expectation values appearing in $\mathrm{tr}[\rho_j \mathcal{L}(O_i)]$

have been calculated leaving the expansion coefficients as the only unknown parameters.

Following the steps above, a complete characterization of the underlying Hamiltonian and dissipative dynamics of the quantum device as given by the Lindbladian is obtained. The two key steps of the protocol are the choice of the set $\{(\rho_j, O_i)\}$ and the polynomial interpolation used to obtain estimates of the time derivatives. Below, we outline the details of both steps and provide rigorous guarantees on the precision of the protocol. Importantly, we show that Lieb-Robinson bounds on the spread of correlations in the system can be used to ensure robust polynomial fitting of the time traces of expectation values allowing for an exponential relaxation of the temporal resolution compared to finite difference methods rendering the protocol feasible for near-term quantum devices.

**Choosing the set of initial states and observables.** The first step in the classical pre-processing is to expand $\mathcal{L}$ in an operator basis

constructed from tensor products of single-qubit Pauli matrices and the identity. The right-hand side (rhs) of the Master equation $\frac{d}{dt}\mathrm{tr}[\rho_j O_i(t)] = \mathrm{tr}[\rho_j \mathcal{L}(O_i)]$ can be expanded as a sum of single Pauli matrices and their products. Our goal is to isolate the unknown expansion coefficients. To this end, we consider an initial state of the form

$$\rho_{k,l}^{(i,j)} = \frac{(I + \sigma_k^{(i)})}{2} \otimes \frac{(I + \sigma_l^{(j)})}{2} \otimes \rho_{n-2}, \qquad (2)$$

where $k, l = x, y, z$. Thus, the $i$'th and $j$'th qubit are prepared in eigenstates of the Pauli matrices $\sigma_k$ and $\sigma_l$ while the state of the remaining $n-2$ qubits, $\rho_{n-2}$, is assumed to be the maximally mixed state.

For a state of the form in Eq. (2) the rhs of the Master equation (see above) can be simplified greatly depending on the choice of the observable $O$. This is due to the properties of the Pauli matrices namely that they have vanishing trace and that

$$\sigma_k \sigma_l = \delta_{kl} I + i\varepsilon_{klp}\sigma_p, \qquad (3)$$

where $\delta_{kl}$ is the Kroenecker delta function and $\varepsilon_{klp}$ is the Levi-Civita symbol. As we show explicitly in the Supplementary Note, if a single qubit Pauli observable, $O = \sigma_l^{(i)}$, is chosen, then only the single qubit terms of the rhs of the Master equation involving the $i$'th qubit will have non-vanishing trace and, using the relation in Eq. (3), the different single qubit Pauli expansion coefficients (the coefficients of terms in the expansion that only involves single qubit Pauli matrices) can be isolated.

After isolating the single qubit expansion coefficients, the coefficients related to two-qubit Pauli terms $(\sigma_i^{(i)} \otimes \sigma_j^{(j)})$ can be isolated by choosing observables of the form $O = \sigma_i^{(i)} \otimes \sigma_j^{(j)}$ in a similar manner. This procedure can be iterated to isolate higher order expansion coefficients by considering observables involving more and and more qubits.

In Supplementary Note, we provide a detailed derivation of how all expansion parameters can be isolated for a general Hamiltonian with terms coupling between two and $k$ qubits and arbitrary single qubit dissipation terms. We note that already for two qubit dissipation terms, deriving linear combinations of initial states and expectation values that allow us to isolate different parameters is quite cumbersome and we do not do this explicitly. However, from a numerical point of view, this is a trivial task. Indeed, as remarked before, each pair of Pauli strings gives us access to a linear equation for the different evolution parameters.

After collecting enough equations to ensure that the linear system is invertible, the precision with which we need to estimate each expectation value to ensure a reliable estimation of the parameters is controlled by the condition number of the matrix describing the system of linear equations. As both estimating the condition number and solving the linear system can be done efficiently, we conclude that estimating dissipative terms acting on a constant number of qubits does not pose a significant challenge from a numerical perspective.

**Robust polynomial interpolation.** As described above, a key step in our learning algorithm is to obtain information about the time-derivatives of observables at $t = 0$. For this, we rely on robust polynomial interpolation. Accordingly, based on expectation values $\langle O_i(t_k)\rangle_j$ for a set of times $t_k$ we want to extract a polynomial $p_{i,j}(t)$ such that we can estimate $\frac{d}{dt}\mathrm{tr}[\rho_j O_i]$ as $\frac{d}{dt}p_{i,j}(t)|_{t=0}$. For this approach to work, we have to be able to control the degree of the polynomial $p_{i,j}(t)$ in order to give an upper bound on the number of sampling points $t_k$ for which we will have to determine $\langle O_i(t_k)\rangle_j$ experimentally. In the

following, we briefly outline how such a guarantee on the degree of $p_{i,j}(t)$ can be obtained and refer to the methods section for a more detailed proof.

Our argument proceeds in two steps: establishing that local expectation values are well approximated by low-degree polynomials and then showing how to robustly extract the derivative of the polynomial from this information. Before we give an overview of the ideas, we will also need to introduce some notation to deal with Lindbladians acting on different parts of the system. We will denote by $\Lambda$ the whole system of qubits and for some subset $B \subset \Lambda$ of qubits, we denote by $\mathcal{L}_B$, the generator truncated to those qubits.

The first step of our proof is to establish that the expectation value $\langle O(t)\rangle$ of a local observable $O$ that evolves under a Lindbladian $\mathcal{L}_B$ restricted to some sub-region $B$ up to some time $t_{max}$, can indeed be approximated up to error $\varepsilon$ by a degree-$d$ polynomial, where $d$ depends linearly on the size of $B$, $t_{max}$ and $\log(\varepsilon^{-1})$.

For the second step of our argument, it remains to show under which circumstances, we can restrict the evolution of the observable $O$, that we identified in the previous step, to a local generator. The main insight here is that for finite range (or sufficiently quickly fast decaying) interactions, the dynamics of any local observable exhibits an effective light cone quantified by a Lieb-Robinson bound (LRB)[43–47]. The LR-bound, in turn, allows us to restrict the Lindbladian on the full system to a generator coupling only systems in the vicinity of the support of $O$, where the size of this shielding region only grows linearly with $t_{max}$.

Bringing these two arguments together, we can first employ the LR-bound to restrict the dynamics to a sub-region around the support of the observable, $O$, and then approximate the corresponding evolution on that finite region up to error $\varepsilon$ by a polynomial of degree $\mathcal{O}[\mathrm{poly}(t_{max}, \log(\varepsilon^{-1}))]$. Now, making use of the techniques from ref. 38, we can extract the first derivative of this polynomial from measurements at $\mathcal{O}[\mathrm{polylog}(\varepsilon^{-1})]$ different times $t_k$. Indeed, in that work, the authors show how to perform polynomial interpolation reliably only given approximations of the values and even under the presence of outliers.

## Robustness of the protocol to experimental errors

In the previous sections, we assumed that we can prepare Pauli eigenstates and measure in Pauli eigenbases and did not consider state preparation and measurement (SPAM) errors. Furthermore, we assumed that the dynamics is described exactly be a generator that is local. Such an idealized scenario rarely comes up in practice and it is important to develop protocols that also work when these conditions are only met approximately. Here, we will provide the main arguments showing that our protocol is indeed robust to such imperfections. We refer to Supplementary Note for the more detailed technical derivations and statements.

First, we consider the setting where the SPAM errors are well-characterized and independent of the basis we prepare and measure. In this case, it is possible to adapt the protocol to incorporate this information without changing its performance significantly. For instance, if the SPAM errors are described by local depolarizing noise with depolarizing probability $p$, we can still recover 2− local Hamiltonians with single qubit noise with a $\mathcal{O}((1-p)^{-4})$ sampling overhead when compared to the noiseless case. In particular, this only depends on the *local* noise rate. We then show that this is the case more generally: if the SPAM errors are not characterized well or are highly dependent on the particular state we are preparing or the basis we are measuring, then once again the effect of SPAM errors will be independent of the system's size and only depend on the precision we wish to obtain.

Besides SPAM errors, our protocol is also robust to perturbations of the generators, even nonlocal ones. More precisely, assume that the true generator is of the form $\mathcal{L} = \mathcal{L}' + \Delta$, where $\mathcal{L}$ is a Lindbladian that

satisfies a LR-bound and $\Delta$ is an arbitrary, potentially global perturbation. A naive bound would imply that for short times, the expectation value of this perturbed evolution could be up to $t\|\Delta\|_{\infty\to\infty}$ away from the unperturbed value, where $\|\Delta\|_{\infty\to\infty} = \sup_{\|X\|_\infty \leq 1} \|\Delta(X)\|_\infty$.

However, we show that only the effect of $\Delta$ on *few-qubit* observables contributes to the bound. A precise statement is given in Thm. 9.1. of the Supplementary Note 9. As an illustrative example, assume that $\Delta$ is a small all-to-all coupling between all pairs of qubits of order $\tau$, e.g.,

$$\Delta(X) = \tau \sum_{i,j=1, i\neq j}^{n} i[\sigma_x^{(i)}\sigma_x^{(j)}, X]. \tag{4}$$

Clearly, such an evolution does not satisfy a LR-bound and $\|\Delta\|_{\infty\to\infty} = \mathcal{O}(\tau n^2)$. However, we show that under such a perturbation, the expectation value is only perturbed by $\mathcal{O}(t\tau n)$, which corresponds to the *local effect of this global perturbation*. To show this, we once again resort to LR bounds and the fact that we only need to measure local observables. Taken together, these results show that our protocol is robust to both SPAM noise and deviations from the assumptions we impose on the generators.

**Learning the structure of the interactions.** So far, we have also assumed that we have knowledge of the interaction graph. However, an astute reader might have remarked that we do not require this explicit knowledge for our protocol to work: indeed, we just use it to decide which parameters to estimate and need to restrict the possible interaction graphs to have an LR bound for the evolution. Thus, a brute-force approach to learning the interaction structure also follows from our results if we wish to estimate all $k$-body couplings of strength at least $\eta$: we just estimate all $k$ body-couplings up to a precision $\eta/4$ and discard all of those that we see are smaller than $\frac{3}{4}\eta$. This can be done with a number of samples that scales like $\mathcal{O}(9^k\eta^{-2}\text{polylog}(\eta^{-1})\log(n))$.

## Numerical examples

To investigate the performance of our protocol for experimentally relevant parameters, we performed numerical simulations of a multi-qubit superconducting device. We consider a system with tunable couplers similar to the Google Sycamore chip[1]. This design relies on a cancellation of the next-next-nearest coupling between two qubits through the direct coupling with a coupler[48,49]. We consider a generic system consisting of a 2D grid of qubits with exchange coupling between nearest neighbors. The dynamics are described through a Lindblad equation with the effective two-qubit Hamiltonian for each neighboring qubit pair $(i,j)$[48,49]

$$H_{ij} = \sum_{k=i,j} \frac{1}{2}\tilde{\omega}_k \sigma_z^{(k)} + \left[\frac{g_i g_j}{\Delta_{ij}} + g_{ij}\right](\sigma_+^{(i)}\sigma_-^{(j)} + \sigma_-^{(i)}\sigma_+^{(j)}) \tag{5}$$

for $i\neq j=1,\ldots,n$ and a dissipation term acting on the $i$'th qubit and having jump operators $\sigma_-^{(i)}, \sigma_+^{(i)}$ (generalized amplitude damping) and $\sigma_z^{(k)}$ (pure dephasing). Here $\tilde{\omega}_k = \omega_k + \frac{g_k^2}{\Delta_k}$ is the Lamb-shifted qubit frequency, $g_i$ is the coupling between the $i$'th qubit and the coupler, and $g_{ij}$ is the direct two-qubit coupling. We have assumed that $\Delta_k = \omega_k - \omega_c < 0$ where $\omega_c$ ($\omega_k$) is the frequency of the coupler ($k$'th qubit) and have defined $1/\Delta_{ij} = (1/\Delta_i + 1/\Delta_j)/2$. By adjusting the frequencies of the coupler and the qubits, the effective qubit-qubit interaction can be canceled up to experimental precision. Typical qubit frequencies are around $5-6$ GHz[1]), while $\Delta_k \sim -1$ GHz, $g_{ij} \sim 10-20$ MHz, and $g_i \sim 100$ MHz[48,49]. In our simulation, we assume that all qubit frequencies and couplings have been characterized up to a precision of 100 kHz using standard characterization techniques[1] and consequently, that all couplers have been tuned off with the same precision i.e., $\frac{g_i g_j}{\Delta_{ij}} + g_{ij} \sim 100$ kHz. Considering a layout of 16 qubits (see below for

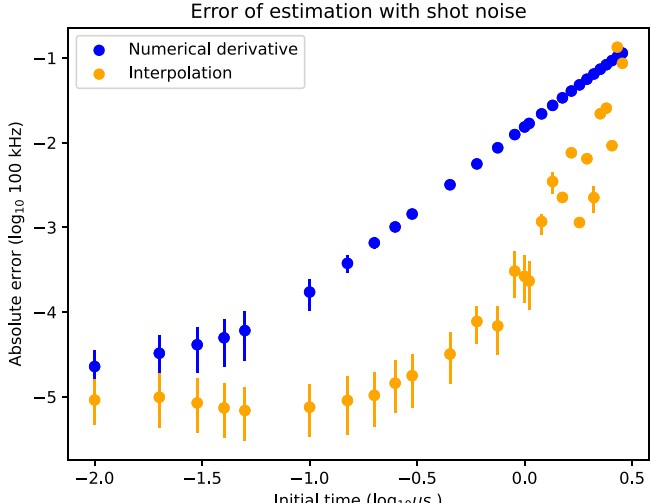

**Fig. 2 | The median quality of recovery of one 2-qubit coupling using interpolation methods and those based on numerical derivatives[30–32] as a function of the initial time.** We assumed that the total time of the experiment is fixed. That is, we let the initial time times the total number of samples for each time step to be a constant ($10^7$ for this plot). We used the expectation value of 40 equally spaced times which had the same distance between each other as the initial time. For each initial time, we simulated 1000 instances of the recovery protocol, always adding shot noise with the same standard deviation to the data. The dots correspond to the median quality of recovery, whereas the lower and upper end correspond to the 25 and 75 percentile. We ran the simulation on a system with 16 qubits.

the interaction graph), we randomly sample all qubit frequencies and qubit-qubit interactions according to Gaussian distributions with zero mean and standard deviation of 100 kHz.

In addition to the Hamiltonian evolution, we also include dissipative dynamics in our numerical simulation. We include quasi-static random frequency shifts of the qubits leading to effective dephasing with a characteristic timescale of $T_2^* \sim 150\,\mu s$ as well as pure dephasing resulting in a transverse relaxation on a timescale $T_2 \sim 60\,\mu s$ representing state of the art coherence times[1,49]. Finally, we include longitudinal relaxation of the qubits through an amplitude damping channel on the time scale of $T_1 \sim 60\,\mu s$. We refer to Supplementary Note 2 for a more detailed discussion and Table III for the sampled parameters of our simulation.

In Fig. 2, we plot the average estimation error as a function of the temporal resolution set by the value of the initial time step, $t_0$. For this plot, we only included the Hamiltonian evolution in the numerical simulation together with quasi-static random frequency shifts of the qubits. This was to lower the run time of the simulation allowing us to investigate the performance for a broad range of initial times. We assumed the total run time of the experiment was fixed such that $t_0 \times S$ is constant, where $S$ is the number of samples. From the figure, we clearly see the improved scaling of our protocol of the estimation error with the time-step size compared to using a finite difference method[30–32]. Besides already performing better at the time resolution for moderate values of the initial time, we see that after a threshold initial time around $10^{-0.7}$, the performance is not limited by the initial time, only the shot noise. In contrast, the finite difference method still requires smaller initial times to improve on the error with the same shot noise.

We also investigated the robustness of our method with respect to shot-noise for a fixed time resolution. For these simulations, we again only included the Hamiltonian evolution together with quasi-static random frequency shifts of the qubits to have a practical run time of the simulation. From Fig. 3, we see that for a fixed time resolution of 30 ns our protocol results in an average estimation error that improves

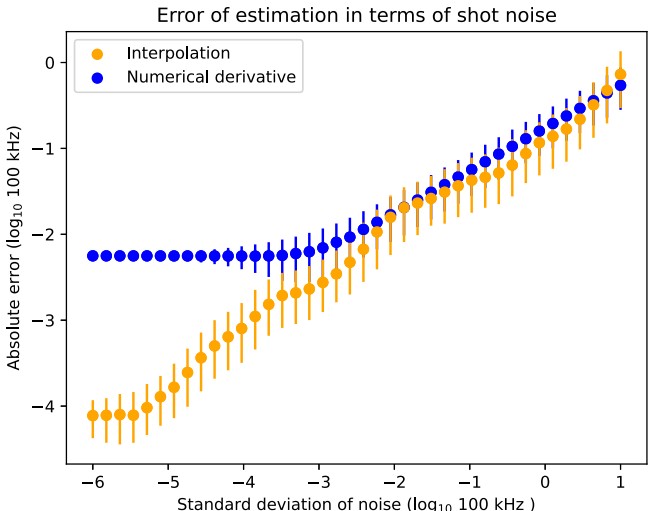

**Fig. 3 | Median quality of recovery of one 2-qubit coupling using interpolation methods and those based on numerical derivatives**[30–32] **as a function of the standard deviation of the shot noise.** The initial time for this estimate is 30 ns and here we also generated 1000 instances of the noise with a given standard deviation. The plot shows the median quality of the recovery and the 25 and 75 percentiles. We see that the quality of the recovery for the interpolation decays approximately linearly with the shot noise, before plateauing at shot noise −6. For the numerical derivative, we see two regimes: first a linear decay of the error until a shot of noise of order $10^{-3}$. After that, the error plateaus and does not improve even with smaller shot noise. This is because for numerical derivative methods, at this point the dominant error source comes from the choice of initial time, whereas for interpolation it is at −6. Importantly, we see that interpolation consistently provides estimates that are no worse than the numerical derivatives method.

linearly with the shot-noise down to an error below $10^{-4}$. This is in contrast to finite difference methods, where the estimation error plateaus around $10^{-3}$ since it becomes limited by the time resolution. This is a clear effect of the exponential improvement of our protocol w.r.t. the time resolution compared to finite difference methods.

Finally, we performed a numerical simulation that included the pure dephasing and amplitude damping noise as described above and estimated the $\sigma_X \sigma_X$ couplings between the qubits. As shown in Fig. 4, we obtain reliable estimates of all 22 couplings demonstrating how our method allows the estimation of specific terms in the Hamiltonian despite the dynamics being governed by the full (dissipative) Lindbladian. For simplicity, we did not explicitly estimate the single qubit Hamiltonian parameters and the Lindbladian decay rates.

For all estimations above, we fitted to degrees 1−7 and picked the one with the smallest average error on the sampled points. In Supplementary Note 6, we give explicit rigorous bounds on how to pick the parameters like the degree for a given desired precision, but we believe that a heuristic approach like the one pursued in the numerics performs well in practice: as long as the interpolating polynomial approximates well the observed points and new points we did not fit to, the degree should be adequate. We note that, although the robust interpolation methods of ref. 38, in principle, require random times, we performed numerical experiments with deterministic times on systems with 16 qubits.

## Discussion

In conclusion, we have proposed a Hamiltonian learning protocol based on robust polynomial interpolation that has rigorous guarantees on the estimation error. Our protocol offers an exponential reduction in the required temporal resolution of the measurements compared to previous methods and a quadratic

reduction in the overall sampling complexity for finite-range interactions. Our protocol only requires the preparation of single qubit states and single qubit measurements in the Pauli bases and is robust to various imperfections such as SPAM errors and Hamiltonian perturbations. This makes it suitable for the characterization of both near-term and future quantum devices.

Furthermore, the recovery of multiple parameters can be highly parallelized by resorting to a variation of classical shadows to quantum channels we improve here.

Our method allows for the characterization of a general local Markovian evolution consisting of a unitary Hamiltonian part and a dissipative part. While we have only explicitly considered single-qubit dissipation here, we believe that our protocol is also valid for general multi-qubit dissipation as outlined above but leave the explicit analysis of this to future work. We have also analyzed the performance of our protocol for algebraically decaying interactions which we believe to be the first results for Hamiltonian learning of such systems. The convergence of our method can be ensured for interactions decaying faster than the dimension of the system. We note, however, that improved bounds on the locality of such systems might improve this result in the future.

## Methods

Here, we detail and formalize our results regarding the estimation error guarantees of our protocol. In particular, we detail the use of Lieb-Robinson bounds on the spread of correlations in the system to bound the error. Furthermore, we outline the shadow tomography method for the parallelization of the measurements.

### Derivative estimation

Define $f(t) = \mathrm{tr}\left[e^{t\mathcal{L}}(O)\rho\right]$ and $\mathcal{L}_B$ to be the Lindbladian truncated to a subregion $B$ of the interaction graph. Our protocol consists of first estimating $f(t_i)$ up to an error $\mathcal{O}(\epsilon)$ for random times $t_1, ..., t_m$. The curve of $f(t)$ is then fitted to a low-degree polynomial $p$, and $p'(0)$ is taken as an estimate for the derivative $f'(0) = \mathrm{tr}[\mathcal{L}(O)\rho]$. Below we prove the accuracy and robustness of this method. The first step is Theorem 1, which establishes under what conditions $f(t)$ is indeed well-approximated by a low-degree polynomial.

**Theorem 1.** Let $\mathcal{L}_\Lambda$ be a local Lindbladian on a $D$-dimensional lattice $\Lambda$. Moreover, let $t_{\max}, \epsilon > 0$ and $O_Y$ be a 2-qubit observable, supported on some region $Y \in \Lambda$, such that $\|O_Y\| \leq 1$, holds. Then there is a polynomial $p$ of degree

$$d = \mathcal{O}\left[\mathrm{poly}(t_{\max}, \log(\epsilon^{-1}))\right], \tag{6}$$

such that for all $0 \leq t \leq t_{\max}$:

$$\left|\mathrm{tr}\left[e^{t\mathcal{L}_\Lambda}(O_Y)\rho\right] - p(t)\right| \leq \epsilon, \tag{7}$$

and $p'(0) = \mathrm{tr}\left[\mathcal{L}_\Lambda(O_Y)\rho\right]$, holds.

The main technical tool required for the proof are LRB[43–47], which ascertain that the dynamics of local observables under a time evolution with a local Lindbladian have an effective lightcone. More precisely, we need that for regions $Y \subset B$ we have

$$\|\left(e^{t\mathcal{L}_B} - e^{t\mathcal{L}_\Lambda}\right)(O_Y)\| \leq c_1 \exp(-\mu \, \mathrm{dist}(Y, \Lambda\backslash\{B\}))(e^{vt} - 1), \tag{8}$$

to hold for constants $c_1, \mu$ and $v$, where dist() is the distance in the graph.

From the LRB we can show that the dynamics is well-approximated by a low-degree polynomial. We leave the details of the proof to Supplementary Note 3 and only discuss the main steps here. The general idea of going from the LRB to the low-degree

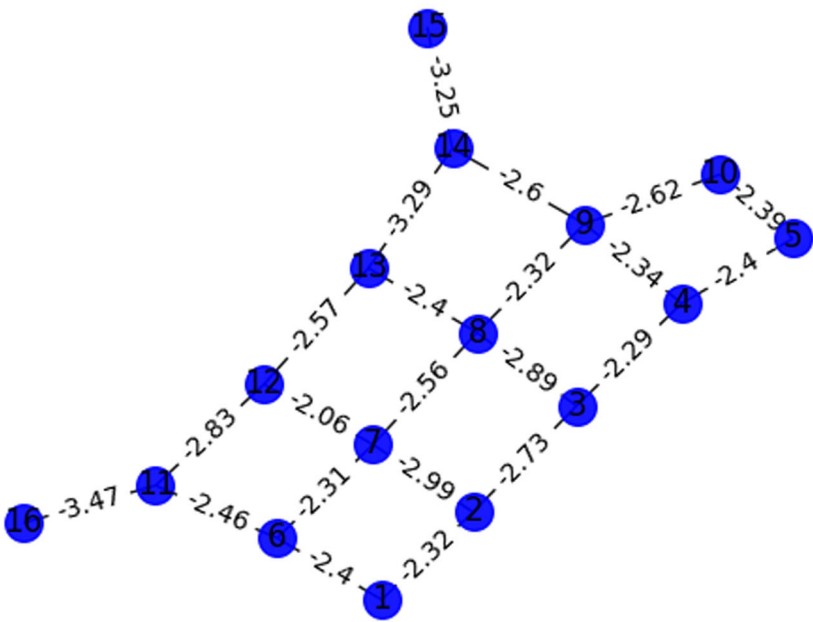

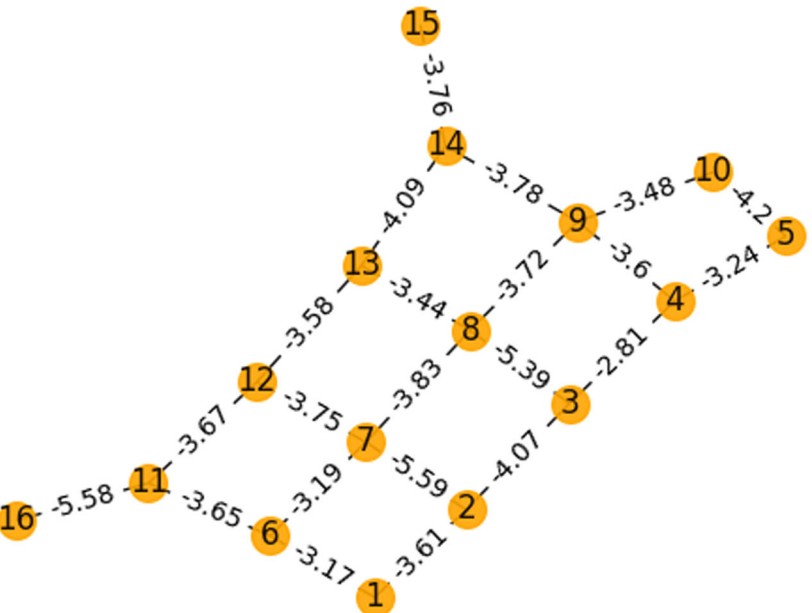

**Fig. 4 | Error in the recovery of $\sigma_X\sigma_X$ couplings of a quantum system with a geometry similar to the Sycamore processor using numerical derivatives and interpolation.** Note that while we only plot the estimation of the Hamiltonian couplings, the numerical simulation included the full Lindbladian including both dephasing due to quasi-static random frequency shifts of the qubits, pure dephasing and amplitude damping noise. The initial time for each coupling was 0.1 μs in the simulation. Note that interpolation consistently outperforms numerical derivatives, sometimes by several orders of magnitude. We chose the time steps and the number of samples to compare both methods, as in Fig. 2.

polynomial is to truncate the Taylor series of the evolution under $\mathcal{L}_B$ for $B$ large enough and take that as the approximating polynomial. As the derivatives of the evolution under $\mathcal{L}_B$ only scale with the size of the region $B$, this allows us to show that the Taylor series converges quickly. To simplify the presentation we did not give explicit numerical constants in the bounds, but in Section A of the Supplementary Note 7 we explicitly determine the constants for the polynomial approximation.

Now that we have concluded that the expectation value is well-approximated by a small degree polynomial, we continue to show that we can reliably extract the derivative from approximations of the expectation values for different $t$. This is formally stated in the following theorem.

**Theorem 2.** Let $\mathcal{L}$ be a Lindbladian on a $D$-dimensional regular lattice. Suppose we can measure the expectation value of two-body Pauli observables on Pauli eigenstates in the time interval $[t_0, t_{max}]$ under $\mathcal{L}$ for $t_0$ as

$$t_0^{-1} = \mathcal{O}\left[\text{polylog}(\epsilon^{-1})\right] \tag{9}$$

and $t_{max} = 2 + t_0$. Then, measuring the expectation values at

$$m = \mathcal{O}\left[\text{polylog}(\epsilon^{-1})\right] \tag{10}$$

random times up to precision $\mathcal{O}(\epsilon/\text{polylog}(\epsilon^{-1}))$, is sufficient to obtain an estimate of the Lindbladian coefficients $\hat{a}_i$ of $a_i$ satisfying

$$|\hat{a}_i - a_i| = \epsilon. \tag{11}$$

This yields a total sample complexity of $S = \mathcal{O}(\epsilon^{-2}\text{polylog}(\epsilon^{-1}))$.

Importantly, Theorem 2 bypasses both requiring small initial times and $\mathcal{O}(\epsilon^{-4})$ sample complexities.

To go from Thm. 1 to Thm. 2 we first need to establish that we can robustly infer an approximation of $p$ from finite measurement data subject to shot noise. Subsequently, we need to show that it will also allow us to reliably estimate $p'(0)$. Let us start with approximating $p$.

### Robust polynomial interpolation
We will resort to the robust polynomial interpolation methods of[38] to show Thm. 2. We review their methods in more detail in Supplementary Note 6. In our setting, the randomly sampled points $x_i, i \in \overline{1,m}$ correspond to different times $\in [t_0, t_{max}]$ and the $y_i \in \mathbb{R}$ to approximations of the expectation value of the evolution at that time. Furthermore, the $y_i$ satisfy the promise that there exists a polynomial $p$ of degree $d$ and some $\sigma > 0$ such, that

$$y_i = p(x_i) + w_i, \quad |w_i| \le \sigma, \tag{12}$$

hold, for strictly more than half of the $y_i$. The rest might be outliers. In our setting, the magnitude of $\sigma$ corresponds to the amount of shot noise present in the estimates of the expectation values.

The authors of[38] then show that by sampling $m = \mathcal{O}(d\log(d))$ points from the Chebyshev measure on $[t_0, t_{max}]$, a combination of $\ell_1$ and $\ell_\infty$ regression allows us to find a polynomial $\hat{p}$ of degree $d$ that satisfies:

$$\max_{x \in [t_0, t_{max}]} |p(x) - \hat{p}(x)| = \mathcal{O}(\sigma). \tag{13}$$

Although the details of the $\ell_1$ and $\ell_\infty$ interpolation are more involved and described in the Supplementary Note 4, a rough simplification of the procedure is the following. First, we find a polynomial $p_1$ of degree $d$ that minimizes $\sum_i |p_1(x_i) - y_i|$. After finding $p_1$ we compute the polynomial $p_\infty$ that minimizes $\max |p_\infty(x_i) - (y_i - p_1(x_1))|$. We then output $\hat{p} = p_1 + p_\infty$ as our guess polynomial. Note that finding both $p_1$

and $p_\infty$ can be cast as linear programs and thus can be solved efficiently[50].

By combining this result with Thm. 1, we robustly extract a polynomial that approximates the curve $t \mapsto \text{tr}\left[e^{t\mathcal{L}}(O_Y)\rho\right]$ up to $\mathcal{O}(\epsilon)$ for $t \in [t_0, t_{max}]$. Indeed, we only need to estimate the expectation value $f(t_i)$ up to $\epsilon$ for enough $t_i$ and run the polynomial interpolation.

Note that Eq. (13) only allows us to conclude that $p - \hat{p}$ is small. However, we are ultimately interested in the curve's derivative at $t = 0$, as the derivative contains information about the parameters of the evolution. For arbitrary smooth functions, two functions being close on an interval does not imply that their derivatives are close as well. Fortunately, for polynomials the picture is simpler. By the Thm. 1 one has to estimate the first derivative of a polynomial at $t = 0$ but not of the actual function. A classical result from approximation theory, Markov brother's inequality[51], allows us to quantify the deviation of the derivatives given a bound on the degree and a bound like Eq. (13). Putting these observations together, we arrive at Thm. 2. The details of the proof are given in the Supplementary Note 4.

### Generalizations of Thm. 2
We also generalize Thm. 2 in two directions. First, we extend the results to interactions acting on $k$ qubits instead of two. As long as the noise is constrained to act on one qubit and $k = \mathcal{O}(1)$, this generalization is straightforward. Indeed, we only need to measure an observable that has the same support as the Pauli string and does not commute with it, as it is then always possible to find a product initial state that isolates the parameter. Generalizing to noise acting on more than one qubit makes it more difficult to isolate the parameters of the evolution as described in the main text. In that case, it then becomes necessary to solve a system of linear equations that couples different parameters. Although our method still applies, analyzing this scenario would require picking the observables and initial states in a way that the system of equations is well-conditioned and we will not discuss this case in detail here.

Second, another important generalization is to go beyond short-range systems. Although we have only stated our results for short-range systems in Thm. 2, our techniques apply to certain long-range systems. As this generalization is more technical, we leave the details to the Supplementary Note 4 and constrain ourselves to discussing how the statement of Thm. 2 changes for more general interactions.

Only one aspect of the previous discussion changes significantly for long-range interactions: how the r.h.s. of Eq. (8) generalizes. More precisely, let us assume that for some injective function $h : \mathbb{R} \to \mathbb{R}$ with $h(x) = o(1)$, we have

$$\| (e^{t\mathcal{L}_B} - e^{t\mathcal{L}_\Lambda})(O_Y) \| \le$$
$$h(\text{dist}(Y, V \backslash \{B\}))(e^{vt} - 1). \tag{14}$$

For instance, for short-range or exponentially decaying interactions, $h$ will be an exponentially decaying function. Then we can restate Thm. 2 in terms of $h^{-1}$. As we show in Thm. 6.1. of the Supplementary Note 6, for a precision parameter $\epsilon > 0$ and evolution on a $D$-dimensional lattice, assume that we pick the initial time as

$$t_0 = \mathcal{O}\left[\left(h^{-1}\left(\frac{\epsilon}{2(e^{2.5v}-1)}\right)^D \log(\epsilon^{-1})\right)^{-2}\right]. \tag{15}$$

Furthermore, assume that we estimate the expectation value of local observables up to precision $\mathcal{O}(\epsilon)$ at $\tilde{\mathcal{O}}[(h^{-1}(\frac{\epsilon}{2(e^{2.5v}-1)})^D \log(\epsilon^{-1}))]$ points. Then we can estimate each parameter up to an error of

$$\mathcal{O}\left[\epsilon\left(h^{-1}\left(\frac{\epsilon}{2(e^{2.5v}-1)}\right)^D \log(\epsilon^{-1})\right)^2\right], \tag{16}$$

through, the same procedure as in the local case. Note that the error in Eq. (16) only tends to 0 as $\epsilon \to 0$ if $h^{-1}(\frac{\epsilon}{2(e^{2.5v}-1)})^D \log(\epsilon^{-1}) = o(\epsilon^{-1})$, holds, i.e., the function $h$ must decay fast enough. In the Supplementary Note 5, we discuss examples of systems with algebraically decaying interactions for which this is satisfied. For instance, for potentials that decay like $r^{-\alpha}$ with $\alpha > 5D - 1$ we obtain that $h^{-1}(\epsilon) = \mathcal{O}(\epsilon^{-\frac{1}{\alpha-3D}})$, holds. We summarize the resulting resources in the Supplementary Table 6.

The message of bounds like (16) is that it is still possible to obtain bounds on the error independent of the system's size beyond short-range systems. However, this comes at the expense of requiring higher precision and sampling from more points.

Another important observation is that the assumption that we know the structure of the interactions exactly is not required. Indeed, our method is robust to Hamiltonian perturbations of the model as long as the resulting evolution still satisfies a LR bound. For instance, suppose that there actually is a non-negligible interaction between qubits $i$ and $j$ that is not accounted by our model. As long as the resulting time evolution still satisfies a LR bound, our results still hold. As the linear equation to isolate any parameter is independent of that parameter, we can still apply our techniques in this setting.

### Parallelizing the measurements

To parallelize the measurement procedure and ensure that we can obtain experimental data to estimate all parameters simultaneously, we resort to a classical shadow process tomography method. Although some papers in the literature already discussed classical shadows for process tomography[39,40], we present a simplified and streamlined proof that also gives an improved sample complexity for the observables relevant to this work in the Supplementary Note.

More precisely, we show that given a quantum channel $\Phi$, Pauli strings $P_a^1, \ldots, P_a^{K_1}$ that differ from the identity on at most $\omega_a$ sites and Pauli strings $P_b^1, \ldots P_b^{K_2}$ that differ from the identity on at most $\omega_b$ sites, it is possible to obtain estimates $\hat{e}_{m,l}$ of $2^{-n} \text{tr}\left[ P_a^m \Phi(P_b^l) \right]$ satisfying

$$|2^{-n} \text{tr}\left[ P_a^m \Phi(P_b^l) \right] - \hat{e}_{m,l}| \le \epsilon \qquad (17)$$

for all $m, l$ with probability at least $1 - \delta$ from

$$\mathcal{O}(3^{\omega_a + \omega_b} \epsilon^{-2} \log(K_1 K_2 \delta^{-1})) \qquad (18)$$

samples. More precisely, the protocol of shadow process tomography requires preparing Eq. (18) many different random initial product Pauli eigenstates and measuring them in random Pauli bases. This makes it feasible to implement it in the near-term. We discuss it in more detail in the Supplementary Note 7, as this protocol may be of interest beyond the problem at hand.

The shadow process tomography protocol is ideally suited for our Hamiltonian learning protocol. Indeed, note that to learn $k$-body interactions, we only required the preparation of initial states $\rho_l$ that differ from the maximally mixed state on $k$ qubits and measure Pauli strings supported on at most $k$ qubits. Furthermore, for a system of $n$ qubits in total, there are at most $16^k \binom{n}{k} \le 16^k n^k$ such states or Pauli strings. We conclude that we can estimate all required expectation values for a given time step using

$$\mathcal{O}(9^k \epsilon^{-2} k \log(n \delta^{-1})) \qquad (19)$$

samples. As our protocol requires estimating expectation values at a total of $\text{polylog}(\epsilon^{-1})$ time steps, we can gather the data required to recover all the $\mathcal{O}(n)$ parameters of the evolution from $\mathcal{O}(\epsilon^{-2} \text{polylog}(n, \epsilon^{-1}))$ samples through the shadow process tomography protocol whenever $k = \mathcal{O}(1)$.

## Data availability

The data generated and analyzed during the current study are available from the authors upon request.

## Code availability

The computer codes developed and tested by the authors, and the input files used for producing the presented data, are available upon request. The current version of the codes is not designed for broad distribution and requires substantial hardware resources for reproducing the presented results. Correct installation of the current version of the codes, preparation of correct input files, and correct analysis of the results require substantial expertise, and may need separate instructions in each specific case.

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

## Acknowledgements

D.S.F. was supported by VILLUM FONDEN via the QMATH Centre of Excellence under Grant No. 10059 and the European Research Council (Grant agreement No. 818761). This work is part of HQI initiative (www.hqi.fr) and DSF was supported by France 2030 under the French National Research Agency award number "ANR-22-PNCQ-0002". A.H.W. thanks the VILLUM FONDEN for its support with a Villum Young Investigator Grant (Grant No. 25452). J.B. acknowledges support from The AWS Quantum Discovery Fund at the Harvard Quantum Initiative and funding from the NWO Gravitation Program Quantum Software Consortium. V.V.D. work is a part of the research program NWO QuTech Physics Funding (QTECH, program 172) with project number 16QTECH02, which is (partly) financed by the Dutch Research Council (NWO); the work was partially supported by the Kavli Institute of Nanoscience Delft. L.A.M. was supported by the Netherlands Organisation for Scientific Research (NWO/OCW), as part of the Quantum Software Consortium program (project number 024.003.037/3368).

## Author contributions

D.S.F., A.H.W., and J.B. conceived the original project. D.S.F., A.H.W., and L.M. conducted the analytical derivations with input from V.V.D. and J.B. The software for the many-qubit simulations was written and tested by V.V.D. L.M. and D.S.F. performed the numerical analysis with input from A.H.W. and J.B. All authors contributed to the discussion of the results and writing of the manuscript.

## Competing interests

The authors declare no competing interests.
