## [Peer Review File · Nature Communications]

Efficient and robust estimation of many-qubit HamiltoniansREVIEWER COMMENTS

Reviewer #1 (Remarks to the Author):

Disclosure: I previously reviewed this paper for the QIP conference. The paper is mostly unchanged, and some of the same comments remain. However, whereas my QIP review put a large emphasis on the theoretical aspects of this paper, my review for this journal puts more emphasis on the practical advantages of the algorithm developed by the authors.

This paper proposes a new algorithm for learning Hamiltonians and Lindbladians from short-time measurements. While previous works mostly focused on estimating the derivatives of local observables from measurements at a fixed time point, the authors suggest to estimate these derivatives by measuring the observables at multiple time points, and utilizing state-of-the-art methods in robust polynomial interpolation. The authors arm their algorithm with theoretical guarantees for its asymptotic sample complexity, which is optimal up to polylogarithmic factors. They benchmark it in simulations of a realistic, experimentally-relevant scenario. Their simulations show a superior performance for their algorithm compared to the approach based on numerical derivatives.

The algorithm developed by the authors seems like a very good practical and experimentally-relevant approach for learning time-independent Hamiltonians.

Asymptotically, a similar scaling can be achieved in theory using either high-order numerical derivatives as in Refs. [30-32], or high-order series expansion as in Ref. [29]. However, I believe the current approach benefits from the advantages of both approaches, yielding superior performance in many scenarios. Compared to the numerical derivatives, the robust polynomial interpolation allows to use data from longer measurement times without incurring systematic errors in the derivative estimation. This allows to improve accuracy and reduce the number of required samples in experimental scenarios. Compared to the high-order series expansion, the classical run-time of the current algorithm seems much more feasible: whereas the series expansion requires fitting a high-degree multivariate polynomial, which becomes computationally daunting at high orders, the current work only requires fitting single-variable polynomials. Moreover, the theoretical analysis of Hamiltonian learning from real-time dynamics is more thorough and complete in the current

work compared to Refs. [29-32].

The only factor which slightly limits the practicality of the algorithm as it is presented is the lack of a method for determining the optimal degree for the polynomial interpolation used for derivative estimation. If I understand correctly, in the simulations the authors used polynomials of different degrees, and chose the degree which minimized the error in the estimated Hamiltonian. However, the error in the estimated Hamiltonian is not available in experiments. I believe the issue of the optimal degree in experimental scenarios can be solved heuristically or numerically. For example, one could use some of the experimental data as 'test data' for the interpolation; or choose the degree that provided an estimated Hamiltonian which best explains the full set of training data. Given such a solution, I believe this method has a high potential for being implemented in future experiments.

Minor comments:

- Some details of the numerical simulations were not clear to me:

o I understand that $t_0 * S$, the initial time t_0 times the number of samples S for each time step, is fixed in Fig. 2. How many different time steps were sampled, and at what intervals?

Alternatively, what were the final time T , and the total number of samples?

o How were observables sampled in the numerical derivative approach: was the total number of samples kept fixed compared to the interpolation (i.e. more samples concentrated at a single time step)? This seems a reasonable thing to do given that measurement and reset times often dominate experiment times, but other interpretations may be that the total evolution time was kept fixed, or the number of samples per time point.

o Was the observable at time '0' assumed to be known, or was it measured as well as part of the total sample budget? (although not necessarily within the scope of this work, this may be interesting in terms of the robustness of the methods to SPAM error)

o Were the same conditions regarding number of samples used also in Fig. 4?

- Regarding the parallelization of measurement settings ('shadow process tomography'):

o How is the parallelization of measurement settings consistent with the isolation of

parameters achieved by initializing qubits to a fully mixed state?

o In particular, is the parallelization of measurements consistent with the robustness to Hamiltonian perturbations (pg. 8)? Are there any assumptions on the locality or magnitude of the unmodeled terms? It seems that parallelizing measurements across regions which were not known to be coupled could be a hassle.

o How does the parallelization of measurements in this paper compare to Ref. [32]?

- The authors write that the average estimation error in the experiments improves linearly with the shot noise, but it seems to improve by 4 orders of magnitude over 6 orders of magnitude of the shot noise; is that expected due to the polylogarithmic factors?

- Some of the notations are used before they are defined (or far apart from their definition without a reference), such as: 'Lv', 'L_Gamma', 'Y', the Hamiltonian terms 'a_alpha' and the Lieb-Robinson velocity 'v'. Also, it seems 'G' is sometimes used to denote the Lindbladian instead of 'L'. Some examples:

o In the statement of Theorem V.1 in the main text, it is implied (but not stated) that 'Y' is the support of the observable O_Y ; the notations G and L are used interchangeably for the Lindbladian; and I am not sure the region 'V' was defined.

o In the statement of Theorem V.2 in the main text, I did not find the definition for the notation $a_{\{\alpha_1, \alpha_2\}}$.

o In Theorem IV.1, the notations G and L are used interchangeably for the Lindbladian in the theorem statement, and it would be useful to remind the reader the notation 'v' refers to the Lieb-Robinson velocity.

Reviewer #2 (Remarks to the Author):

The present paper addresses the timely and important topic of learning Hamiltonians or Lindbladians from experimental data. Especially for the characterization of interactions and noise processes in quantum devices, this is a highly relevant task and a prerequisite for accurate theoretical modelling, improvements in experimental design, and eventually the quantification of gate errors. Consequently, Hamiltonian learning has received a lot of (theoretical) attention lately.

The authors present a learning method which is based on the short-time dynamics governed by a Master equation. They exploit that the Master equation reduces to a linear system of equations for the coefficients of the Lindbladian in the Pauli basis when evaluated at time $t=0$. These equations are obtained by suitable choices of initial states and observables which can be taken to factorize.

Then, the only crucial ingredient left is the estimation of the time derivatives of the relevant expectation values at $t=0$. Here, the authors introduce a novel method based on polynomial interpolation which overcomes several shortcomings of previous approaches. To this end, the authors use Lieb-Robinson bounds to show that the time-dependent expectation values are indeed well-described by a low-degree polynomial, and combine this with a recent method for polynomial interpolation which is robust against statistical outliers. Importantly, it is shown that the derivative of the so-obtained polynomial is a good approximation to the exact derivative.

For the estimation of the time-dependent expectation values, the authors propose to use a variant of shadow estimation for Pauli observables. More precisely, shadow estimation in Pauli bases is used to efficiently estimate weight k Pauli observables (with $k=O(1)$) and this is sufficient to estimate the coefficients of k -body terms in the Lindbladian.

In my opinion, the present work is well executed and overcomes several limitations of previous methods. The authors obtain convincing guarantees for their protocol which makes it likely that it can be successfully performed in experiments. Thus, the results of this work are certainly of broad interest to the quantum science community and neighboring fields. The technical contribution consists of a clever and innovative combination of Lieb-Robinson bounds, polynomial interpolation methods, and the use of shadow estimation for expectation values. Because of this I recommend that the paper is accepted in Nature Communications after a minor to moderate revision (see below).

The reason for the revision is first and foremost that the manuscript is in an unpolished state. I think that some parts of the paper need clarification, see below. Moreover, there are plenty of typos, notational inconsistencies, and other minor problems which I tried to list below. Second, when mentioning previous works based on the Master equation approach, the paper "Practical Black Box Hamiltonian Learning" by Gu et al. should be explicitly

discussed. This work also uses polynomial interpolation and achieves a similar scaling in the approximation error.

Comments

- I think the authors should explicitly include the shadow tomography part into their protocol, to make this part as clear as possible and easy to use. Currently, the "Quantum processing" part of the protocol in Sec. II is not the one for which the guarantees are derived. Furthermore, the transition to the discussion of the shadow tomography protocol at the end of page 3 should be improved. To most readers, it is probably not obvious how the shadow tomography protocol is used exactly for the estimation of the expectation values. It might be worthwhile to remind the reader what Φ is in this context, and how the matrix coefficients are combined linearly to obtain the relevant expectation values. The exact usage of shadow tomography could be clarified by writing out an alternative "Quantum processing" step at the end of Sec. II.A.

- I am a bit confused about the details of the numerical simulation.

It seems to me that the shadow estimation part of the protocol is not simulated, instead the expectation values are evaluated directly (and averaged over realizations of the noise), and shot noise is added by hand. Then, I am wondering what "number of samples" means in this context?

- While the big-O notation is fine to give an overview of the scaling, it would be interesting to know the constants for a practical implementation of the protocol. Judging from the derivation in the supplemental material, it should not be too difficult to make this explicit. I would appreciate it if the authors could give an explicit formula (including the dependency on the locality) somewhere, e.g. in the methods section.

- I checked the technical part of the paper and found no (uncorrectable) mistakes.

Minor comments

Main text:

- Fig. 1: "set of initial product states and single qubit"
- 'time trace' is a bit non-standard and not quite self-explanatory. Why not simply 'expectation value'?
- p. 2: "orthogonal set of *traceless* Hermitian operators". Remove 'traceless' (not important here and technically speaking, your definition of Pauli basis includes the identity)
- p. 2: "It is well known that the Master equation [...]" is quite a convoluted sentence. Since this is crucial for the remainder of the paper, this paragraph should be clarified. I think it would also be worthwhile to already mention here that you intend to estimate the time derivative at $t=0$.
- p. 3, Eq. (2): Implicit Einstein summation, consider writing the sum over m explicitly
- the discussion about isolating the expansion coefficients right after is a bit confusing to read. It is formulated as if it would be evident that the choice of states and observables would allow for the isolation. Thus, the reader is stuck at this point trying to figure out what clever two-line reasoning would lead to this conclusion. At this point, the reader has not even seen the full expansion yet. Looking at the supplementary material, the argument is far from straightforward. I think this is simply a wording problem and formulations like "As we show explicitly later, ..." or "It is possible to show ..." or the like would resolve this easily.
- p. 4 "swe"
- p. 5: "demonstrating how our method allows the estimation of specific terms in the Lindbladian despite the dynamics being governed by the full (dissipative) Lindbladian". Should the first 'Lindbladian' be 'Hamiltonian'?
- Theorem V.1: Y, V, \mathcal{G} is not defined
- p. 7 "But they depart ..." I think that's not what the authors want to express here
- p. 7 "for polynomials the situation is much simpler". Consider reminding the reader here that by Thm V.1 one only has to estimate the first derivative of a polynomial at 0 (not of the actual function)
- p. 8 "to acts on 1 qubit"
- p. 8 "precious"

- Inconsistent and partially confusing notation (both within main text and supplementary material as well as in between both texts):
- n vs N for number of qubits (c.f. page 1/2 vs 3)
- indices: (ρ_j, O_i) but also α_i
- σ is once used for single-qubit Paulis, and P for n -qubit Paulis, but in the beginning, the Pauli basis is $\{\sigma_i\}$
- in the supplemental material, coefficients are a (not α) and Pauli indices are given by Greek letters (better convention IMO)
- why P_a and P_b ? a, b are just decorations, not variable indices. P_1, P_2 would make more sense (you even use K_1 and K_2). Even better: Use different variables, why the need for so many indices? E.g. P and \tilde{P} or P and Q
- w vs ω for weight. w_a vs $w(P_a)$ etc
- $\|\cdot\|$ is used for the operator/spectral norm without introduction (and inconsistently with the supplemental material, see below)
- The precise dependency of sampling complexity on different parameters such as locality etc is a bit obscured and can only be found in the very end. Consider mentioning that earlier

Supplemental Material

- Plenty of dead references (??) (e.g. p. 11), plenty of typos (some are listed below)
- p. 12 "Given a system of n qubits on a D , we let ...". I guess Γ denotes the lattice in this context.
- Which norm is used in (69) ? cb norm?
- Thm. IV.1 / Prop. IV.1 / Thm. IV.2 (both): Lindbladian is denoted by \mathcal{G} in the assumptions, but by \mathcal{L} in the statement and elsewhere.
- Thm. IV. 1 "and O_Y and"
- p. 13 "can only increases"
- Induced norms are introduced on p. 13. It's probably a good idea to define the stabilized versions there as well before they are used later (p. 23) (I am not even sure why the stabilized versions are needed, though). The notation for the stabilized norms is also a bit bulky and non-standard. I'd rather use $\|\cdot\|_{\mathrm{cb}}$ and $\|\cdot\|_{\diamond}$ instead.
- Consider to consistently use $\|\cdot\|_{\infty}$ for the operator/spectral norm. Right now,

it is a mix of this notation with $\|\cdot\|$ (without decoration).

- Lemma IV.1 ``define the evolution of the truncated evolution''

- There's a wrong argument in the proof of Lemma IV.1. In particular, Eq. (74) is incorrect, but Eq. (75) is correct. The problem lies in the expression of the derivative of f_B . Clearly, the k -th derivative of $e^{t\mathcal{L}_B}$ is $\mathcal{L}_B^k e^{t\mathcal{L}_B}$ and the t^k in the first equation of the proof is wrong. Then, the bound in Eq. (74) is $g^k |B|^k$. In combination with the Lagrange form of the Taylor series remainder, this gives the correct expression (75) (otherwise, we would have t_{max}^2).

- p. 14 ``procl''

- p. 17, after Lemma V.1: ``CM''

- p. 22 ``where d could refer'' should be dist

- p. 24: Again, notation

- p. 24: Why the t in Φ_t ?

- p. 24: the notion of ``overlap'' is a bit bulky. Wouldn't it be simpler to say that P_a is diagonal in the measured basis or not?

- p. 26, Eq. (139): $P(B \text{ and } S \text{ overlap})$ should be removed on the LHS

- p. 27: ``Pauli states''

Reviewer #3 (Remarks to the Author):

Summary: The authors provide a method for estimating the parameters of a Lindbladian on an underlying multi-qubit quantum system. This imports the usual assumptions of Markovian and time-homogenous dynamics. They show how specific parameters of the Lindbladian can be isolated through appropriate choice of initial separable states and final few-qubit measurements. Measuring expectation values at various times allows these isolated parameters to be determined through polynomial regression techniques. They address how to limit the degree of the fitting polynomial in order to avoid over-fitting. Importantly they also show how to overcome problems with time-resolution intervals of previous protocols, if one is able to make assumptions as the finite range of interactions in the underlying mechanics. Shadow-tomography techniques are employed to parallelize the measurements. The protocol assumes knowledge of the general interaction graph of the device, i.e. which qubits are coupled with each other, but no knowledge of the strength or

form of such interactions.

This is a well written paper with many nice aspects to it. Much of the previous literature concentrates on Hamiltonians and the extension to Lindbladians is timely and welcome. The numerics are experimentally motivated which is a nice touch, although as far as we can tell the implementation of the numerics makes the same underlying assumptions as the model in question. The working through of the relevant input/output pairs in the supplemental material is also welcome and greatly enhances the ease with which the paper is understood and the useability of its ideas.

There are, however, a couple of points we would like clarified before recommending the paper for publication in a journal such as Nature Communications.

The first concern relates to the effect imperfect state preparation and measurement will have on the protocol. If we were to attempt to use the method in an experimental setting it is very likely there will be errors in the preparation of states and the measurement of observables. The protocol involves very precise state/observable pairs that eliminate most of the variables to be determined. By eliminating the effect of most of the parameters in the Lindbladian (through this clever choice of input/output), then it becomes possible to fit the few parameters that survive. The concern here stems from the fact that noisy state preparation/measurement of observables will mean that the other parameters (and there could be a lot of them) are not eliminated from the calculation. This does not appear to be addressed by the authors and could have a huge impact on the ability of the protocol to be used in an experimental context. The word "robust" in the title, refers to the polynomial fitting procedure (so, we presume robust to shot noise). (As a side note if the protocol is not robust to SPAM errors then perhaps the title should be revisited). Perhaps there is some way to quantify the error caused by imprecise preparation/measurement or show how the protocol is robust to that error too.

The second concern relates to what happens if the structure is "guessed" wrongly and the actual underlying structure is more complex. This is certainly of practical concern, cross-talk and parasitic couplings between qubits is one of the main noise sources

seen in current devices. Does the model being created degrade gracefully? In classical, say, graphical model literature, there is much effort in showing that if the graphical model does not fully capture all the relations in the underlying process being modelled, the model degrades in a graceful way. Perhaps the authors can comment on the effect of there being additional unaccounted parameters (for instance, if certain qubits are linked in the “actual” Linbladian describing the system e.g. distant qubits linked because of frequency crowding, but are not included in the ansatz. Is the model still approximately correct, or are “all bets off”? Perhaps there are simulations that have been done where the underlying simulation assumes dynamics more complicated than that being fit?

There are a few other points we wish to raise, in no particular order:

1. In column 2, it is not obvious how one goes from measuring things $O(\epsilon)$ apart and a precision of $O(\epsilon^{2-2})$ to a complexity of $O(\epsilon^{-4-4})$ (as opposed to $O(\epsilon^{-3-3})$)
2. Talking about all the parameters of local n -qubit Linbladian in $O(\epsilon^{-2-2}\text{polylog}(n, \epsilon^{-1-1}))$ is a bit confusing when the authors can be a bit more precise. We understand the point is the logarithmic scaling in system size, but hiding the locality parameter behind the big-O notation just confuses. We would suggest it might be better to talk about k -local n qubit Linbladians with a complexity of (we believe) $O(9^{k-2}\epsilon^{-2-2}k\text{polylog}(n, \epsilon^{-1-1}))$. Then (as in the conclusion) when $k=O(1)$ the quoted complexity follows.
3. Perhaps the authors could confirm if there is a hidden exponential scaling, in that as n increases ϵ will have to get smaller and smaller (ϵ being additive here), in order for any useful information to be learned.
4. We are a little bit uncomfortable with the authors' bold assertions that previous works were incorrect.

The essence of “shadow tomography” is that $\approx 2n$ parameters can be measured simultaneously. This is now well known (shadow tomography papers, Cotler and Wilzcek - Quantum Overlapping Tomography, Flammia & Wallman - Efficient estimation of Pauli channels.) Therefore if the number of parameters to be measured can be reduced to have

an exponential component no larger than $O(2^{2n})$ the problem can be made efficient. This has been shown for instance in ref 27 where these techniques are used to reduce the number of measurements required for Hamiltonian measurement to something that scales with the log of system size (as in this manuscript). Obviously if the number of parameters scale exponentially at a rate greater than $O(2^{2n})$ there is still an exponential number of measurements involved. This is true for Pauli channels as well as “full” process tomography. To the point in question, the claim that the authors in (their refs) [39 and 40] miss the exponential pre-factor and are therefore wrong, needs more back up if it is to be made. A perusal of ref [40] shows that gives the scaling that you would expect given the above description. They very quickly start to talk about reduced k-qubit matrices which they wouldn’t need to do if they had missed the scaling. Their full protocols do indeed have a pre-factor that is exponential in the system size. I haven’t followed through the maths. The authors may wish to expressly set out where they believe the other papers go wrong, or just go ahead and present their own proof, which is a fine proof. I would recommend the latter (remembering to fix up the claim at the end of page 1). Finally on this point the authors talk about it being ‘shadow process tomography’. It isn’t. Process tomography involves determining all of the coefficients of a process matrix. Here there is an underlying simplification that this is represented by a Lindbladian with a number of parameters at most $O(2^{2n})$. The authors need to be more careful with their claims especially end of page 8 and page 9.

5. In the caption figure 3 the claim is the interpolation consistently provides better estimates, but in the graph shown the error bars appear to overlap after about -3 on the x-axis. Why better and not no worse than? Also on this point (figure 2 as well) in the text it refers to “our protocol” and the “finite difference” method, but the keys in the graphs are “Interpolation” and “Numerical derivative”. This just puts an unnecessary cognitive load on the reader.

6. There are a number of typos. I know the editors of the journal will go through checking for this, but the manuscript would benefit from close checking. Below I set out a few I have noticed:

- page 4, first column “such that we can estimate”
- page 5, the “Finally, we performed a full numerical simulation including also the pure” might be a bit clumsy, “Finally we performed a numerical simulation that included .. ” seems

better.

- caption figure 4 "usign"
- Theorem V.1 - I have lost track of what G is.
- page 8 top, "noise is constrained to acts on 1 qubit". Should it be act?
- page 8 third paragraph "precious" → "previous"

Referee 1

Disclosure: I previously reviewed this paper for the QIP conference. The paper is mostly unchanged, and some of the same comments remain. However, whereas my QIP review put a large emphasis on the theoretical aspects of this paper, my review for this journal puts more emphasis on the practical advantages of the algorithm developed by the authors.

This paper proposes a new algorithm for learning Hamiltonians and Lindbladians from short-time measurements. While previous works mostly focused on estimating the derivatives of local observables from measurements at a fixed time point, the authors suggest to estimate these derivatives by measuring the observables at multiple time points, and utilizing state-of-the-art methods in robust polynomial interpolation. The authors arm their algorithm with theoretical guarantees for its asymptotic sample complexity, which is optimal up to polylogarithmic factors. They benchmark it in simulations of a realistic, experimentally-relevant scenario. Their simulations show a superior performance for their algorithm compared to the approach based on numerical derivatives.

The algorithm developed by the authors seems like a very good practical and experimentally-relevant approach for learning time-independent Hamiltonians. Asymptotically, a similar scaling can be achieved in theory using either high-order numerical derivatives as in Refs. [30-32], or high-order series expansion as in Ref. [29]. However, I believe the current approach benefits from the advantages of both approaches, yielding superior performance in many scenarios. Compared to the numerical derivatives, the robust polynomial interpolation allows to use data from longer measurement times without incurring systematic errors in the derivative estimation. This allows to improve accuracy and reduce the number of required samples in experimental scenarios. Compared to the high-order series expansion, the classical run-time of the current algorithm seems much more feasible: whereas the series expansion requires fitting a high-degree multivariate polynomial, which becomes computationally daunting at high orders, the current work only requires fitting single-variable polynomials. Moreover, the theoretical analysis of Hamiltonian learning from real-time dynamics is more thorough and complete in the current work compared to Refs. [29-32].

Response: We are very happy that the referee values our work and acknowledges its practical value for experimentalists and its improvement on state-of-the-art.

The only factor which slightly limits the practicality of the algorithm as it is presented is the lack of a method for determining the optimal degree for the polynomial interpolation used for derivative estimation. If I understand correctly, in the simulations the authors used polynomials of different degrees, and chose the degree which minimized the error in the estimated Hamiltonian. However, the error in the estimated Hamiltonian is not available in experiments. I believe

the issue of the optimal degree in experimental scenarios can be solved heuristically or numerically. For example, one could use some of the experimental data as ‘test data’ for the interpolation; or choose the degree that provided an estimated Hamiltonian which best explains the full set of training data. Given such a solution, I believe this method has a high potential for being implemented in future experiments.

Response: We agree with the referee that a better discussion of how to pick the degree of the fitting polynomials would be helpful to the readers. We have therefore added a new subsection in the supplemental material (Certifying the degree of the polynomial in the fitting) that details this. In our numerical simulations, we did indeed take a heuristic approach as suggested by the referee. We increased the degree of the polynomial until the fitting curve hit all points up to an error of order ϵ . That is, we *did not* pick a degree that delivered good results for the (in practice unknown) parameter, but rather one that provided a good fit with the simulated data points. One additional heuristic one can follow is to pick a few additional points, as suggested by the referee, and check that they are still within a tolerated window. Depending on this, we accept or reject the corresponding estimate. We expect this heuristic strategy to perform very well in practice, as suggested by our numerics.

Besides suggesting this heuristic strategy that bears little overhead, we also propose a rigorous strategy in the supplemental material. In this version of the paper we give explicit bounds on the degree of the polynomial in Sec. VII.A. These can be used in the estimation.

Minor comments:

- Some details of the numerical simulations were not clear to me:

o I understand that $t_0 * S$, the initial time t_0 times the number of samples S for each time step, is fixed in Fig. 2. How many different time steps were sampled, and at what intervals? Alternatively, what were the final time T , and the total number of samples?

Response: We used 40 time steps, all equally spaced and with a spacing that is the same as the initial time. We have now added this information to the caption of Fig. 2.

o How were observables sampled in the numerical derivative approach: was the total number of samples kept fixed compared to the interpolation (i.e. more samples concentrated at a single time step)? This seems a reasonable thing to do given that measurement and reset times often dominate experiment times, but other interpretations may be that the total evolution time was kept fixed, or the number of samples per time point.

Response: We used the same observables for the derivatives and the interpolation method. Note that the way we isolate the parameters is independent of how we obtain the estimates of the derivatives, so there is no a-priori reason to

pick different observables for the methods. And yes, we kept the total number of samples fixed for both methods, but then for the derivatives method dedicated all the sample budget to the estimation of the expectation values at t_0 .

o Was the observable at time ‘0’ assumed to be known, or was it measured as well as part of the total sample budget? (although not necessarily within the scope of this work, this may be interesting in terms of the robustness of the methods to SPAM error)

Response: When performing the numerics we did not put the $t = 0$ point in the interpolation. So we are not using this information for the fitting. However, we are using it later solve the system of equations. We also note that we have now added a Sec. IX in the supplemental material as well as a subsection II.C in the main manuscript where we explicitly discuss how to adjust the protocol in the presence of SPAM errors so that they are taken into account when we solve the system of equations.

Were the same conditions regarding number of samples used also in Fig. 4?

Response: Yes. We have now also added this information to the figure caption.

- Regarding the parallelization of measurement settings (‘shadow process tomography’):

o How is the parallelization of measurement settings consistent with the isolation of parameters achieved by initializing qubits to a fully mixed state?

Response: In the previous version of the work, we first explained the version of the protocol where the initial state is fixed to isolate a given parameter. If one indeed follows this elementary approach (i.e. prepare the initial state tailored to isolate one parameter), we cannot parallelize the estimation. However, with the randomized measurements and initial state preparation of the shadow protocol, it is possible to estimate in parallel the expectation value of the experiments “as if” we prepared mixed states to isolate the parameters by weighting the outcome statistics appropriately. Thus, the parallelization and isolation of the parameters happens through our postprocessing. We reshaped the main text to reflect this point more accurately.

o In particular, is the parallelization of measurements consistent with the robustness to Hamiltonian perturbations (pg. 8)? Are there any assumptions on the locality or magnitude of the unmodeled terms? It seems that parallelizing measurements across regions which were not known to be coupled could be a hassle.

Response: We thank the referee for pointing out these important questions regarding unknown couplings in the Hamiltonian. We have added a new sections IX-X in the supplementary material as well as subsections II.C-D in the main manuscript to address exactly this and related issues. In summary, the situation is as follows: if we have unmodeled Hamiltonian terms, as long as the true evolution still satisfies a LR-bound, then we can still run our protol, isolate

the parameters and obtain the same recovery guarantee. If we have "rogue" dissipative terms that couple qubits, then we can detect them, as long as the true evolution still satisfies a LR bound. This is discussed in Sec. IX of the supplemental material. Finally, we also consider perturbations that destroy the LR bound, e.g. an all-to-all coupling. In Thm. IX.1, we give precise statements on how large the perturbation can be (depending on the minimal time we want to use), for our protocol to still work. Interestingly, also here we can use the fact that the evolution is a perturbation of an evolution that satisfies a LR bound and get a perturbation bound that is quadratically better than what one would expect naively.

o How does the parallelization of measurements in this paper compare to Ref. [32]?

Response: In Ref. [32], the authors come up with a highly effective way of parallelizing the measurements for a *given initial state*. But, in principle, several different initial states might still be required to estimate various parameters. In contrast, our approach parallelizes *both the initial states and the measurements*. This comes at the cost that we can only estimate expectation values for initial states that are maximally mixed except at a small number of qubits. However, as such initial states suffice for our purposes, we can parallelize both state initialization and measurements.

- The authors write that the average estimation error in the experiments improves linearly with the shot noise, but it seems to improve by 4 orders of magnitude over 6 orders of magnitude of the shot noise; is that expected due to the polylogarithmic factors?

Response: The polylogarithmic factors certainly play a role here, however, we expect that another effect is dominant in the plot of Fig. 3: the choice of initial time. Note that the initial time is fixed for that plot, and that the quality of the recovery becomes constant around shot noise of order 10^{-5} . We expect that, in order to reach an even higher accuracy, it is necessary to go to smaller initial times.

Some of the notations are used before they are defined (or far apart from their definition without a reference), such as: 'Lv', 'L.Gamma', 'Y', the Hamiltonian terms 'a_alpha' and the Lieb-Robinson velocity 'v'. Also, it seems 'G' is sometimes used to denote the Lindbladian instead of 'L'. Some examples:

In the statement of Theorem V.1 in the main text, it is implied (but not stated) that 'Y' is the support of the observable O_Y ; the notations G and L are used interchangeably for the Lindbladian; and I am not sure the region 'V' was defined.

In the statement of Theorem V.2 in the main text, I did not find the definition for the notation $a_{\{\alpha_1, \alpha_2\}}$.

In Theorem IV.1, the notations G and L are used interchangeably for the Lind-

bladian in the theorem statement, and it would be useful to remind the reader the notation refers to the Lieb-Robinson velocity.

Response: We thank the referee for pointing out all of these typos and missing definitions. We have carefully gone through the manuscript and corrected these errors.

Referee 2

The present paper addresses the timely and important topic of learning Hamiltonians or Lindbladians from experimental data. Especially for the characterization of interactions and noise processes in quantum devices, this is a highly relevant task and a prerequisite for accurate theoretical modelling, improvements in experimental design, and eventually the quantification of gate errors. Consequently, Hamiltonian learning has received a lot of (theoretical) attention lately.

The authors present a learning method which is based on the short-time dynamics governed by a Master equation. They exploit that the Master equation reduces to a linear system of equations for the coefficients of the Lindbladian in the Pauli basis when evaluated at time $t=0$. These equations are obtained by suitable choices of initial states and observables which can be taken to factorize. Then, the only crucial ingredient left is the estimation of the time derivatives of the relevant expectation values at $t=0$. Here, the authors introduce a novel method based on polynomial interpolation which overcomes several shortcomings of previous approaches. To this end, the authors use Lieb-Robinson bounds to show that the time-dependent expectation values are indeed well-described by a low-degree polynomial, and combine this with a recent method for polynomial interpolation which is robust against statistical outliers. Importantly, it is shown that the derivative of the so-obtained polynomial is a good approximation to the exact derivative. For the estimation of the time-dependent expectation values, the authors propose to use a variant of shadow estimation for Pauli observables. More precisely, shadow estimation in Pauli bases is used to efficiently estimate weight k Pauli observables (with $k=O(1)$) and this is sufficient to estimate the coefficients of k -body terms in the Lindbladian.

In my opinion, the present work is well executed and overcomes several limitations of previous methods. The authors obtain convincing guarantees for their protocol which makes it likely that it can be successfully performed in experiments. Thus, the results of this work are certainly of broad interest to the quantum science community and neighboring fields. The technical contribution consists of a clever and innovative combination of Lieb-Robinson bounds, polynomial interpolation methods, and the use of shadow estimation for expectation values. Because of this I recommend that the paper is accepted in Nature Communications after a minor to moderate revision (see below).

Response: We are very happy that the referee finds our work well executed and significant. As detailed below, we have substantially improved the manuscript based on the comments of the referee to make it suitable for publication in Nature Communications.

The reason for the revision is first and foremost that the manuscript is in an unpolished state. I think that some parts of the paper need clarification, see below. Moreover, there are plenty of typos, notational inconsistencies, and

other minor problems which I tried to list below. Second, when mentioning previous works based on the Master equation approach, the paper “Practical Black Box Hamiltonian Learning” by Gu et al. should be explicitly discussed. This work also uses polynomial interpolation and achieves a similar scaling in the approximation error.

Response: We have substantially revised the manuscript and polished it. Furthermore, we have also added a reference to the work by Gu et al. We do note, however, that our work appeared before theirs.

- I think the authors should explicitly include the shadow tomography part into their protocol, to make this part as clear as possible and easy to use. Currently, the “Quantum processing” part of the protocol in Sec. II is not the one for which the guarantees are derived. Furthermore, the transition to the discussion of the shadow tomography protocol at the end of page 3 should be improved. To most readers, it is probably not obvious how the shadow tomography protocol is used exactly for the estimation of the expectation values. It might be worthwhile to remind the reader what Φ is in this context, and how the matrix coefficients are combined linearly to obtain the relevant expectation values. The exact usage of shadow tomography could be clarified by writing out an alternative “Quantum processing” step at the end of Sec. II.A.

Response: We agree with the referee that the parallel estimation procedure based on shadow tomography required a more significant integration in the manuscript. Following the advice of the referee, we now describe this in detail as part of the description of the quantum processing step in Section II.A. In addition, we have updated figure 1 to also encompass this parallel approach.

- I am a bit confused about the details of the numerical simulation. It seems to me that the shadow estimation part of the protocol is not simulated, instead the expectation values are evaluated directly (and averaged over realizations of the noise), and shot noise is added by hand. Then, I am wondering what “number of samples” means in this context?

Response: The referee is correct that we have simulated the expectation values directly. The reason we decided for this approach instead of implementing the shadows protocol is that we considered relatively large systems (up to 15 qubits). Generally speaking, computing expectation values is significantly simpler than obtaining samples from a state when performing classical simulations, so obtaining the shadows would have been too expensive from a computational point of view. Of course, when considering real devices, obtaining the samples is straightforward.

To add the shot noise, we used the fact that, in the limit of a large number of samples, by the central limit theorem, the shot-noise will be well-approximated by a normal distribution with mean 0 and standard deviation $1/\sqrt{N}$, where N is the number of samples. Thus, we set a value for the number of samples and added a sample from the corresponding normal variable to the result. This way

we can reproduce the effect of shot noise in the data.

- While the big-O notation is fine to give an overview of the scaling, it would be interesting to know the constants for a practical implementation of the protocol. Judging from the derivation in the supplemental material, it should not be too difficult to make this explicit. I would appreciate it if the authors could give an explicit formula (including the dependency on the locality) somewhere, e.g. in the methods section.

Response: We agree with the referee that providing as much information as possible about the scaling constants is valuable- To this end, we have now added a new Sec. VII.A in the supplemental material, where we discuss various models that details the scaling constants of the Lieb-Robinson bounds used in our derivations.

Response on minor comments: The referee had a very thorough list of typos/inconsistent notation and missing definitions in both the main manuscript and the supplemental material. For brevity, we have not included the entire list in this response but we thank the referee for pointing our all of these small errors. We have carefully gone through the manuscript and corrected these in addition to general polishing of the manuscript.

Referee 3

Summary: The authors provide a method for estimating the parameters of a Lindbladian on an underlying multi-qubit quantum system. This imports the usual assumptions of Markovian and time-homogeneous dynamics. They show how specific parameters of the Lindbladian can be isolated through appropriate choice of initial separable states and final few-qubit measurements. Measuring expectation values at various times allows these isolated parameters to be determined through polynomial regression techniques. They address how to limit the degree of the fitting polynomial in order to avoid over-fitting. Importantly they also show how to overcome problems with time-resolution intervals of previous protocols, if one is able to make assumptions as the finite range of interactions in the underlying mechanics. Shadow-tomography techniques are employed to parallelize the measurements. The protocol assumes knowledge of the general interaction graph of the device, i.e. which qubits are coupled with each other, but no knowledge of the strength or form of such interactions.

This is a well written paper with many nice aspects to it. Much of the previous literature concentrates on Hamiltonians and the extension to Lindbladians is timely and welcome. The numerics are experimentally motivated which is a nice touch, although as far as we can tell the implementation of the numerics makes the same underlying assumptions as the model in question. The working through of the relevant input/output pairs in the supplemental material is also welcome and greatly enhances the ease with which the paper is understood and the useability of its ideas.

Response: We thank the referee for these very positive remarks about our work.

There are, however, a couple of points we would like clarified before recommending the paper for publication in a journal such as Nature Communications.

The first concern relates to the effect imperfect state preparation and measurement will have on the protocol. If we were to attempt to use the method in an experimental setting it is very likely there will be errors in the preparation of states and the measurement of observables. The protocol involves very precise state/observable pairs that eliminate most of the variables to be determined. By eliminating the effect of most of the parameters in the Lindbladian (through this clever choice of input/output), then it becomes possible to fit the few parameters that survive. The concern here stems from the fact that noisy state preparation/measurement of observables will mean that the other parameters (and there could be a lot of them) are not eliminated from the calculation. This does not appear to be addressed by the authors and could have a huge impact on the ability of the protocol to be used in an experimental context. The word “robust” in the title, refers to the polynomial fitting procedure (so, we presume robust to shot noise). (As a side note if the protocol is not robust to SPAM errors then perhaps the title should be revisited). Perhaps there is some way to quantify the error caused by imprecise preparation/measurement or show how

the protocol is robust to that error too.

Response: We agree with the referee that we should have been more explicit about the various ways our protocol is robust to various imperfections. We have now added a new Section IX in the supplemental material as well as subsections II.B-C in the main manuscript that discuss various aspects of the robustness of our protocol.

First, we describe a general theorem that shows that if the true evolution deviates by $\sim \epsilon$ from the true one (including spam errors) for all time steps samples, then our protocol can still recover the parameters up to ϵ . Thus, any sort of perturbation of magnitude at most ϵ will not hugely impact the protocol. Then we discuss the effect of SPAM errors more explicitly in Sec. IX.B-C. Here we have results on two complementary regimes: first, we show that if the quantum channels that captures the SPAM errors are well-characterized, this information can be easily incorporated into the protocol and, for local noise models, not significantly affect the performance. On the other hand, if we do not know the SPAM errors, we show that only local SPAM errors affect the performance. For example, we show that if each read-out and state preparation fails with probability p , then the total error this will cause in our estimates will be of order $\mathcal{O}(p \log(\epsilon^{-1}))$, which is independent of the system size. Thus, as long as local SPAM errors are smaller than the desired accuracy, the recovery still works.

In section IX.A, we also discuss one further form of robustness, namely that to nonlocal interactions. We show that even if we have a small all-to-all coupling present in the interactions on top of the local dynamics, the expectation values are affected by how this perturbation acts locally. Thus, the protocol is also robust to deviations from the model we assume (local interactions).

The second concern relates to what happens if the structure is “guessed” wrongly and the actual underlying structure is more complex. This is certainly of practical concern, cross-talk and parasitic couplings between qubits is one of the main noise sources seen in current devices. Does the model being created degrade gracefully? In classical, say, graphical model literature, there is much effort in showing that if the graphical model does not fully capture all the relations in the underlying process being modelled, the model degrades in a graceful way. Perhaps the authors can comment on the effect of there being additional unaccounted parameters (for instance, if certain qubits are linked in the “actual” Lindbladian describing the system e.g. distant qubits linked because of frequency crowding, but are not included in the ansatz. Is the model still approximately correct, or are “all bets off”? Perhaps there are simulations that have been done where the underlying simulation assumes dynamics more complicated than that being fit?

Response: We thank the referee for raising this important question. Besides the discussion of perturbations in section IX.A mentioned above, we have also added a new section X in the supplemental material together with a subsection II.D in the main manuscript that address this. In summary, we believe there

are two regimes to be considered here: if the "rogue" coupling does not perturb the dynamics enough to destroy the LR bound, then we can essentially argue as before, the conclusions remain unchanged. We show that, as long as we have a LR bound and a lower bound on the couplings, we do not need to know the exact coupling structure. On top of that, we show that even if the perturbation is "very global" and destroys the LR bounds, our protocol is still quadratically more robust than one would expect naively. This is the content of Theorem IX.1, where we give explicit bounds on the magnitude of the errors incurred by perturbations that potentially destroy the LR bound. In short, we believe that, our new results show that our scheme is robust to various forms of perturbations of the underlying model. Furthermore, we added a result on learning the structure of the interactions in Section X detailing how we can learn the interaction graph and detect rogue couplings. This way, we do not even require the exact graph as an input to the problem.

There are a few other points we wish to raise, in no particular order:

In column 2, it is not obvious how one goes from measuring things $O(\epsilon)$ apart and a precision of $O(\epsilon^2)$ to a complexity of $O(\epsilon^{-4})$ (as opposed to $O(\epsilon^{-3})$)

Response: The ϵ apart refers to the time of the measurement, that is, for such methods we need to measure small time differences or only after evolving for a small time. This does not influence the sample complexity, but might be challenging in some experimental setups. However, to achieve a precision of ϵ^2 , it is necessary to measure ϵ^{-4} time. So the ϵ^{-4} comes from the requirement on the precision.

2. Talking about all the parameters of local n-qubit Lindbladian in $O(\epsilon^{-2}\text{polylog}(n, \epsilon^{-1}))$ is a bit confusing when the authors can be a bit more precise. We understand the point is the logarithmic scaling in system size, but hiding the locality parameter behind the big-O notation just confuses. We would suggest it might be better to talk about k-local n qubit Lindbladians with a complexity of (we believe) $O(9k\epsilon^{-2}k\text{polylog}(n, \epsilon^{-1}))$. Then (as in the conclusion) when $k=O(1)$ the quoted complexity follows.

Response: We thank the referee for making this valid point and we agree that it is important to be transparent about all scalings. We added the locality dependency (dependency on k) also in the introduction.

3. Perhaps the authors could confirm if there is a hidden exponential scaling, in that as n increases ϵ will have to get smaller and smaller (ϵ being additive here), in order for any useful information to be learned.

Response: We are not sure we understand what is meant by the "hidden exponential scaling". How precisely we need to learn the parameters of course depends on the desired application and this can be optimized in various ways.

For the sake of providing clarity, let us, however, take a rather strong definition of "useful information": let us suppose that we wish to be sure that the dynamics

generated by the simulator is ϵ close in trace distance to a Lindbladian we wish to simulate for all times $\leq t_{\max}$. As illustrated by the perturbation bound we use in the new section on robustness of the protocol (Eq. Theorem IX.1), it suffices to ensure that the difference between the actual generator and the desired generator (Δ) in our notation, satisfies $\|\Delta\|t_{\max} \leq \epsilon$. For local dynamics, the number of terms in Δ will be polynomial in n . Thus, if we learn each term up to a precision $\epsilon/\text{poly}(n)t_{\max}$, we can ensure that the underlying dynamics will only differ by ϵ for all time range. In short, if the dynamics is local and we only wish to simulate it for polynomial times up to an error that is polynomially small, this can all be achieved with a number of samples that is polynomial. Thus, even in this very strong sense of recovery that certainly yields useful information, we would only need a polynomial number of samples. Of course, if we desire exponential precision, go to exponential times or have dynamics with exponentially many parameters, then some form of exponential scaling is unavoidable. But we believe that this is not the case at hand and thus believe that there is no hidden exponential scaling.

4. We are a little bit uncomfortable with the authors' bold assertions that previous works were incorrect.

The essence of "shadow tomography" is that $2n$ parameters can be measured simultaneously. This is now well known (shadow tomography papers, Cotler and Wilzcek - Quantum Overlapping Tomography, Flammia and Wallman - Efficient estimation of Pauli channels.) Therefore if the number of parameters to be measured can be reduced to have an exponential component no larger than $O(2n)$ the problem can be made efficient. This has been shown for instance in ref 27 where these techniques are used to reduce the number of measurements required for Hamiltonian measurement to something that scales with the log of system size (as in this manuscript). Obviously if the number of parameters scale exponentially at a rate greater than $O(2n)$ there is still an exponential number of measurements involved. This is true for Pauli channels as well as "full" process tomography. To the point in question, the claim that the authors in (their refs) [39 and 40] miss the exponential pre-factor and are therefore wrong, needs more back up if it is to be made. A perusal of ref [40] shows that gives the scaling that you would expect given the above description. They very quickly start to talk about reduced k -qubit matrices which they wouldnt need to do if they had missed the scaling. Their full protocols do indeed have a pre-factor that is exponential in the system size. I haven't followed through the maths. The authors may wish to expressly set out where they believe the other papers go wrong, or just go ahead and present their own proof, which is a fine proof. I would recommend the latter (remembering to fix up the claim at the end of page 1). Finally on this point the authors talk about it being 'shadow process tomography'. It isn't. Process tomography involves determining all of the coefficients of a process matrix. Here there is an underlying simplification that this is represented by a Lindbladian with a number of parameters at most $O(2n)$. The authors need to be more careful with their claims especially end of

page 8 and page 9.

Response: We thank the referee for pointing this out. We believe that there was some confusion on this point from our side and we apologize for this. First, we believe that it is important to point out that one previous proof was indeed incorrect. As clear evidence of this, we can look at how the statement of Theorem 2 evolved in the article "Shadow process tomography of quantum channels". If we look at version 2 (available <https://arxiv.org/pdf/2110.03629v2.pdf>), we see that N has no exponential dependence in n . If we then look at version 3 (available <https://arxiv.org/pdf/2110.03629v3.pdf>), we see that the authors then added a 2^n factor. Unfortunately, this is still incorrect. The correct factor is 4^n , as stated in the fourth version. Thus, the current arXiv version indeed has a correct version of the statement. The latest version was, however, only uploaded on the 11.04.23, months after we posted our work. However, we must also admit that we were not careful enough when analyzing the work of Levy, Luo and Clark. Their conventions and notation are slightly different than ours which, when combined with the other result that was indeed wrong, lead to the hasty judgement it was also incorrect.

We agree with the referee that our initial claims were too bold and we have therefore toned them down in the updated manuscript. We have however, still kept our own proof. The reason for this is two-fold: first, we believe it is good to have a proof that is more self-contained and does not rely on shadow norms. Second, the fact that we only consider Pauli matrices allows us to obtain a better sample complexity compared to previous results (3^w instead of 4^w).

5. In the caption figure 3 the claim is the interpolation consistently provides better estimates, but in the graph shown the error bars appear to overlap after about -3 on the x-axis. Why better and not no worse than? Also on this point (figure 2 as well) in the text it refers to "our protocol" and the "finite difference" method, but the keys in the graphs are "Interpolation" and "Numerical derivative". This just puts an unnecessary cognitive load on the reader.

Response: We agree with the referee that the suggested alterations are suitable, as the previous version of the text was confusing or imprecise. We have changed the captions of both Fig. 2 and 3 accordingly.

6. There are a number of typos. I know the editors of the journal will go through checking for this, but the manuscript would benefit from close checking. Below I set out a few I have noticed:...

Response: We thank the referee for pointing out all of these typos. We have carefully gone through the manuscript and corrected these errors.

REVIEWERS' COMMENTS

Reviewer #1 (Remarks to the Author):

My concerns have been addressed by the revisions. The revised version expands the applicability of the results to a variety of realistic settings, and I believe it is fit for publication as is.

Minor comments on the revisions and response:

- The fact that the interaction graph can be learned whenever it satisfies a LR bound is quite important. It is interesting to compare to learning of classical Ising / graphical models, where the learning of the graph can be performed efficiently whenever the width of the graph, i.e. the l_1 norm of interactions on each site, is bounded by a constant - see the following:

<https://arxiv.org/abs/1706.06274>

The bounded graph width is indeed a prerequisite for the LR bounds to hold.

- SPAM errors: I understand that characterized SPAM errors are easy to overcome in the current setting, but uncharacterized spam errors are more challenge. In particular, if I understand correctly, generic uncharacterized SPAM errors of magnitude τ can limit the reconstruction of each term approximately to the same precision τ (is that correct?). Although beyond the scope of this manuscript, I wonder if this can be improved in theory or at least in practice, given that often SPAM errors can be forced to act as depolarizing in many realistic scenarios (e.g. through twirling).

- Regarding the process shadow tomography relative to ref. 32 - indeed that reference first demonstrates it for measurements of a given state, but then it applies the same process on initial states as well, leading to the same sample complexity as described here if I am not mistaken. Perhaps an important difference is that the current approach is armed with bounds and performance guarantees that apply to the non-geometrically-local case.

- Perhaps I have not read it too much carefully, but can't $t_0 = d_{\text{exp}}^2$ get prohibitively large in eq. 126, such that measurements restricted to the interval between t_0 and $t_{\text{max}} = 2 + t_0$

provide little information?

Reviewer #2 (Remarks to the Author):

I think that my critique has been appropriately addressed in the revised manuscript. Moreover, the other referees raised some important questions on the robustness against perturbations of the Lindbladian and SPAM robustness. In my opinion, both have been reasonably discussed in the revision. Thus, I recommend acceptance of the revised manuscript.

Reviewer #3 (Remarks to the Author):

I thank the authors for their considered and comprehensive review of my comments and concerns.

I am particularly grateful for the addition of sections addressing the robustness of their methods to SPAM errors and 'rogue' couplings. I appreciate that this was no small task.

I am very happy to recommend the publication of the manuscript.

(Just for the sake of completeness I spotted a missing space after the reference to Section IV - in the text after equation 13, and after the reference to Section VI - in the text after equation 15).

Reviewer 1:

My concerns have been addressed by the revisions. The revised version expands the applicability of the results to a variety of realistic settings, and I believe it is fit for publication as is.

Reply: We are happy to see that we addressed the concerns of the referee and that they are now recommending acceptance.

Minor comments on the revisions and response: - The fact that the interaction graph can be learned whenever it satisfies a LR bound is quite important. It is interesting to compare to learning of classical Ising / graphical models, where the learning of the graph can be performed efficiently whenever the width of the graph, i.e. the l_1 norm of interactions on each site, is bounded by a constant - see the following: <https://arxiv.org/abs/1706.06274> The bounded graph width is indeed a prerequisite for the LR bounds to hold.

Reply: We agree that the fact that our protocol can learn the graph from the data given that it satisfies a LR bound is important and gives another reason why our protocol is advantageous compared to others in the literature. We chose not to give this point further space and emphasis considering the already considerable length of our manuscript.

The potential connection to the classical problem of learning the Ising model is indeed very interesting. Unfortunately, to the best of our knowledge, bounded width is not sufficient (only necessary) for LR. So further research might be required to elevate our results to interactions of bounded width.

- SPAM errors: I understand that characterized SPAM errors are easy to overcome in the current setting, but uncharacterized spam errors are more challenge. In particular, if I understand correctly, generic uncharacterized SPAM errors of magnitude τ can limit the reconstruction of each term approximately to the same precision τ (is that correct?). Although beyond the scope of this manuscript, I wonder if this can be improved in theory or at least in practice, given that often SPAM errors can be forced to act as depolarizing in many realistic scenarios (e.g. through twirling).

Reply: The referee is correct wrt. to the effect of non-characterized SPAM errors. We agree that twirling methods are something natural to try in our setting to bring the noise into a normal form. We hope to address this point in future work.

- Regarding the process shadow tomography relative to ref. 32 - indeed that reference first demonstrates it for measurements of a given state, but then it applies the same process on initial states as well, leading to the same sample complexity as described here if I am not mistaken. Perhaps an important dif-

ference is that the current approach is armed with bounds and performance guarantees that apply to the non-geometrically-local case.

Reply: Yes, we agree that it would be interesting to deepen our understanding of the differences between the two approaches, but the locality aspect is likely key as pointed out by the referee.

- Perhaps I have not read it too much carefully, but can't $t_0 = d_e x p^2$ get prohibitively large in eq. 126, such that measurements restricted to the interval between t_0 and $t_m a x = 2 + t_0$ provide little information?

Reply: As long as the precision is polynomial in the system's size and the lattice's dimension is constant, then t_0 will remain polylogarithmic in system size, and thus not get prohibitively large.

Reviewer 2

I think that my critique has been appropriately addressed in the revised manuscript. Moreover, the other referees raised some important questions on the robustness against perturbations of the Lindbladian and SPAM robustness. In my opinion, both have been reasonably discussed in the revision. Thus, I recommend acceptance of the revised manuscript.

Reply: We are happy to see that the referee is satisfied with the changes we implemented and is now recommending acceptance.

Reviewer 3

I thank the authors for their considered and comprehensive review of my comments and concerns.

I am particularly grateful for the addition of sections addressing the robustness of their methods to SPAM errors and 'rogue' couplings. I appreciate that this was no small task.

I am very happy to recommend the publication of the manuscript.

(Just for the sake of completeness I spotted a missing space after the reference to Section IV - in the text after equation 13, and after the reference to Section VI - in the text after equation 15).

Reply: We are happy to see that we were able to address the concerns of the referee and that they now recommend publication. We have corrected the typo, thank you for the careful reading!